# RobustMAD: Evaluating Real-World Robustness of Multimodal Small Language Models for Deployable Anomaly Detection Assistants

**Anushiya Arunan**[1,2]                *anushiya_arunan@mymail.sutd.edu.sg*
**Xin Li**[3]                        *xin019@e.ntu.edu.sg*
**Yan Qin**[4]                       *yan.qin@cqu.edu.cn*
**U-Xuan Tan**[1]                     *uxuan_tan@sutd.edu.sg*
**Nhu Khue Vuong**[2]               *vuong_nhu_khue@a-star.edu.sg*
**Xiaoli Li**[1]                      *xiaoli_li@sutd.edu.sg*
**Chau Yuen**[3,*]                    *chau.yuen@ntu.edu.sg*

[1] *Singapore University of Technology and Design, Singapore*
[2] *Agency for Science, Technology and Research (A\*STAR), Singapore*
[3] *Nanyang Technological University, Singapore*
[4] *Chongqing University, China*

Reviewed on OpenReview: *https://openreview.net/forum?id=skrA9UYNIZ*

**Code:** https://github.com/en-research/RobustMAD
**Project Page:** https://robustmad.github.io/

## Abstract

Multimodal industrial anomaly inspection assistants are a critical component of next-generation smart factories, enabling interactive vision–language–based querying. However, multimodal large language models remain impractical for on-site deployment due to prohibitive computational demands and privacy risks from cloud-based inference. Compact multimodal *small* language models (MSLMs) offer a deployable alternative, yet progress is constrained by the lack of comprehensive robustness analyses and meaningfully challenging benchmarks that reflect real-world industrial conditions. To address this gap, we develop RobustMAD, the first deployment-motivated benchmark, designed to comprehensively evaluate model robustness through diverse open-ended queries spanning object understanding, anomaly detection, unanswerable problems, and visual quality degradations. Contrary to conventional assumptions, top-performing MSLMs exhibit promising capabilities, surprisingly outperforming even the larger GPT-5 Nano. However, they still fall short of safety-critical requirements, and RobustMAD reveals critical robustness gaps that pose operational risks. In particular, three recurring failure modes emerge: (i) fragile multimodal grounding under fine-grained distinctions or degraded visual conditions, (ii) insufficiently comprehensive responses, and (iii) weak logical grounding on unanswerable or ill-posed queries, leading to hallucinated outputs. Grounded in these insights, we provide actionable guidance for the design of next-generation multimodal industrial inspection assistants that leverage their promising competence.

---

[*]Corresponding author

# 1 Introduction

Artificial intelligence agents on embodied edge devices are attracting increasing attention in industrial settings for their potential to amplify human productivity and reduce repetitive manual effort and tedium. Within this context, multimodal visual assistants for industrial anomaly inspection form a critical component of next-generation smart factories (Jiang et al., 2025). Recent advances in foundation multimodal large language models (MLLMs) have dramatically expanded the capabilities of industrial anomaly detection, from rigid, vision-only anomaly detection systems to a more interactive vision and language–based, open-ended querying and inspection reporting. However, MLLMs, especially proprietary models, have limited practical utility in real-world factory environments due to computational resource constraints on edge devices and privacy concerns associated with transmitting sensitive industrial data to cloud-based services.

Given these limitations, there has been a timely shift in research focus toward multimodal ***small*** language models (MSLMs), which are typically at or below 4 billion parameters, often open-sourced, and amenable to local deployment on resource-constrained devices (Jin et al., 2025; Ahmed et al., 2025). Recent work has focused on expanding the foundational capacities of efficient MSLMs, substantially narrowing the performance gap with moderate-scale models across a wide range of standard benchmarks (Yang et al., 2025; Yao et al., 2024; Ahmed et al., 2025). However, strong benchmark performance in controlled evaluations does not necessarily translate to robust performance in practice (Gong et al., 2025). Real-world industrial inspection settings are highly complex, characterized by (i) domain-intensive knowledge requirements, (ii) dynamic variations in image quality, (iii) ill-posed or non-standardized user queries, and (iv) the open-ended nature of responses, where no predefined multiple-choice answers are available. Thus, for real-world robustness, a model must understand the underlying logic of the inspection task and exhibit grounded reasoning across diverse variations of the problem, based on user queries.

**Existing gap.** However, both comprehensive robustness analyses of MSLMs and meaningfully challenging benchmarks for real-world industrial anomaly detection remain severely lacking—marking a significant gap in realizing efficient, multimodal anomaly inspections on-site. Most existing benchmarks instead focus on large models, which are unsuitable for on-device inference in resource-constrained environments, and more critically, do not meaningfully assess model robustness against realistic, diverse, and practical user queries (Jiang et al., 2025). Motivated by this gap, we seek to (i) illuminate how ready current state-of-the-art MSLMs are for deployment as accurate, helpful, and robust visual inspection assistants, and (ii) characterize the critical capabilities and design priorities essential for next-generation MSLMs to achieve real-world robustness. As an initial illustrative peek into the broader robustness challenges, Figure 1 shows how even minor variations in query phrasing, which are common in practical inspection settings, can lead to substantially different model responses and potentially catastrophic consequences in safety-critical product inspections.

**Our work.** We present Robust Multimodal Anomaly Detection (RobustMAD), the first deployment-motivated benchmark for evaluating the robustness of MSLMs in industrial anomaly inspection, a domain that stands to significantly benefit from on-device MSLMs. RobustMAD is explicitly designed to capture core practical challenges, including domain-intensive reasoning, non-standardized and ill-posed user queries, open-ended inspection reasoning, and realistic visual quality variations. Our benchmark comprehensively encompasses both major types of robustness: **knowledge-based robustness**, which evaluates model reasoning under diverse, imperfect, and domain-knowledge–intensive queries, and **visual quality robustness**, which assesses model sensitivity to image-quality perturbations commonly occurring in dynamic assembly lines, such as motion blur and low lighting.

We utilize 410 representative images from the popular MVTec AD (Bergmann et al., 2019) and VisA (Zou et al., 2022) anomaly detection datasets to meticulously curate 4,510 multiple-choice and 7,380 open-ended questions, spanning four knowledge-based robustness categories. These categories are designed to collectively capture the diverse demands of real-world anomaly inspection: *General Object Understanding*, *Stand-alone Anomaly Detection*, *Pair-wise Anomaly Detection*, and *Unanswerable or Ill-posed Query Detection*. With data efficiency in mind, we intentionally prioritize high-quality, meaningfully challenging questions over sheer dataset scale. By systematically covering diverse challenges such as domain knowledge–intensive scientific objects, fine-grained defects, realistic visual quality perturbations, cross-image reasoning, open-ended queries,

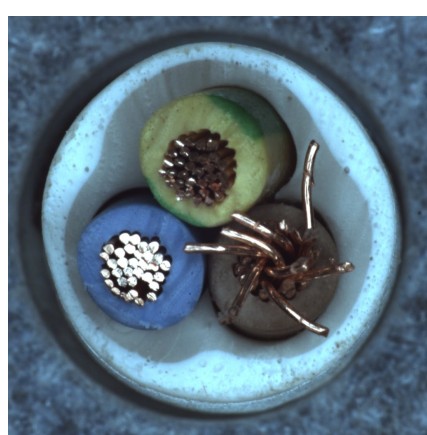

**Query A**

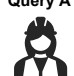

*" Does the object in the image have any defects? If so, describe the defect or anomaly and specify its location (e.g., upper-left, center, bottom-right)."*

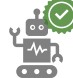

The object shows **exposed, frayed copper wires at the center-right**, indicating a defect in insulation or cable integrity. This anomaly compromises safety and functionality, suggesting improper manufacturing or damage during handling.

**Query B**

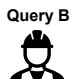

*" Confirm that there are no major defects in the object shown."*

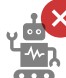

**No major defects observed**. The cable's internal structure shows intact insulation layers and properly bundled conductors. All components appear securely assembled with no visible damage or irregularities. The cross-section is clean and well-defined.

Figure 1: **Fragility of logical grounding in state-of-the-art MSLMs.** While the model correctly identifies exposed wires under a neutral query (Query A), it fails to ground its reasoning and hallucinates a "defect-free" state under a confirmation-seeking query (Query B). This illustrates how practical variations in query phrasing can produce catastrophic false negatives in safety-critical inspections due to robustness gaps.

and notably unanswerable problems, RobustMAD enables, for the first time, a rigorous evaluation of the fragilities and failure modes of modern MSLMs (see Table 1).

With RobustMAD, we conduct an extensive empirical evaluation of state-of-the-art MSLMs, together with moderate-scale open-source and proprietary models as references. Though top-performing MSLMs exhibit surprisingly promising baseline competence—exceeding even larger models such as Phi-4-Multimodal-Instruct (6B) and proprietary GPT-5 Nano—we uncover critical robustness gaps that have been largely obscured by standard benchmarks and aggregate accuracy metrics. Particularly, three recurring failure modes emerge under open-ended evaluation: (i) fragile multimodal grounding, in which reasoning beyond macro visual attributes fails under fine-grained distinctions or degraded visual conditions; (ii) insufficiently comprehensive responses, lacking the precision and explanatory detail required by industrial inspection standards; and (iii) weak logical grounding on unanswerable or ill-posed queries, resulting in erroneous, hallucinated answers that disregard missing evidence. To enable standardized assessment of such behaviors, we design a multi-dimensional, user-centered scoring scheme for our LLM judge, which evaluates MSLM responses against real-world requirements of technical accuracy, comprehensiveness, relevance, and style and clarity. Together, the RobustMAD benchmark and our findings offer a timely evaluation of current MSLM readiness for industrial anomaly inspection, reveal critical failure modes that pose significant operational risks, and provide practical guidance for designing next-generation multimodal inspection assistants.

Our key contributions are summarized as follows:

1. We present, to the best of our knowledge, the first comprehensive evaluation of MSLM robustness for multimodal industrial anomaly inspection, a domain where MSLMs are critical for on-device deployment but remain largely unexplored.

2. We develop RobustMAD, a rigorous benchmark for systematically evaluating model robustness through diverse open-ended inspection queries spanning object understanding, anomaly detection, unanswerable problems, and visual-quality degradations, thus revealing deeper failure modes that standard benchmarks fundamentally cannot capture.

3. We design a multi-dimensional, user-centered scoring scheme to holistically evaluate MSLMs on technical accuracy, comprehensiveness, relevance, and clarity—enabling standardized and quantitative assessment of open-ended responses. The human-verified ground truth answers used for evaluation are also released to support future research in the broader community.

Table 1: Comparison of proposed RobustMAD with traditional industrial anomaly detection (IAD) datasets and representative multimodal evaluation benchmarks.

| Feature | Real-IAD (Wang et al., 2024) | Adversarial VQA (Li et al., 2021) | Visual Robust VQA (Ishmam et al., 2025) | MMAD (Jiang et al., 2025) | RobustMAD (Ours) |
|---|---|---|---|---|---|
| Domain | IAD | General | General | IAD | IAD |
| Input modality | Image only | Image, Text | Image, Text | Image, Text | Image, Text |
| Domain knowledge-intensive? | ✓ | ✗ | ✗ | ✓ | ✓ |
| Cross-image reasoning? | ✗ | ✗ | ✗ | ✓ | ✓ |
| Complex open-ended queries? | ✗ | Short-form OE | Short-form OE | ✗ | ✓ |
| Realistic negative queries? | ✗ | ✗ | ✗ | ✗ | ✓ |
| Visual quality robustness? | ✓ | ✗ | ✓ | ✗ | ✓ |
| Evaluation metric | AUROC | Accuracy | Accuracy | Accuracy | LLM Judge w/ multi-dimensional score |
| Analysis coverage | Vision-only | MLLMs | MLLMs | MLLMs | MSLMs (w/ reference MLLMs) |

4. We uncover surprising strengths of even generalist MSLMs, challenging conventional assumptions about the benefits of model scaling, while also revealing critical failure modes largely overlooked in prior analyses. Grounded in these findings, we offer actionable guidance for designing next-generation multimodal inspection assistants.

The rest of the paper is organized as follows. Section 2 reviews related work and highlights the gaps we aim to address. Section 3 presents our RobustMAD benchmark dataset, including the data creation pipeline and key statistics. Section 4 discusses experimental results, identifies failure modes, and provides practical guidance for next-generation MSLMs. Finally, Section 5 concludes the paper and outlines directions for future work.

## 2 Related Works

We organize prior works into three areas most relevant to this study: (i) benchmarks for multimodal small language models, (ii) multimodal industrial anomaly detection, and (iii) robustness-focused visual question answering. Together, the works reveal that comprehensive robustness analyses of MSLMs and meaningfully challenging benchmarks for evaluating real-world industrial anomaly detection are severely lacking, highlighting a significant gap in realizing efficient, on-site multimodal anomaly inspections.

### 2.1 Multimodal Small Language Model Benchmarks

Compared to their larger-scale counterparts, comprehensive analyses of small language models, and particularly multimodal small language models, remain relatively scarce. SLM-Bench (Pham et al., 2025) is the first comprehensive benchmark for small language models (SLMs, not multimodal), evaluating their performance across multiple natural language tasks, general knowledge domains, hardware configurations, and environmental impacts. A similar, though less comprehensive, evaluation has been conducted in surveys (Jin et al., 2025; Ahmed et al., 2025) to benchmark MSLM performance on standard vision-language benchmark datasets (Goyal et al., 2017; Fu et al., 2025; Lu et al., 2024; 2022). However, none of these studies examine robustness of MSLMs in depth, nor do they address industrial anomaly detection, a domain that can significantly benefit from MSLMs deployed on-device.

### 2.2 Multimodal Industrial Anomaly Detection Benchmark Datasets

Typically, industrial anomaly detection has been treated as a computer vision problem, focusing on object- or pixel-level defect detection (Pemula et al., 2025; Jayasekara et al., 2023; Zhang et al., 2022). Recent attempts to incorporate multimodal vision–language inputs are still in their infancy and broadly fall into two directions: works that evaluate the anomaly inspection capabilities of specific state-of-the-art MLLMs (e.g., GPT-4V) through qualitative case studies (Cao et al., 2023), and approaches that customize MLLMs

for anomaly detection, either through carefully crafted prompting strategies (Xu et al., 2024) or fine-tuning on anomaly detection datasets (Gu et al., 2024). Against these fragmented efforts, the MMAD benchmark (Jiang et al., 2025) represents the first systematic evaluation of multiple MLLMs on generalized industrial anomaly inspection tasks. However, despite its large scale, MMAD relies on a restricted multiple-choice format that neither reflects realistic open-ended inspection queries nor probes model robustness in depth. As a result, its practical utility for assessing MSLMs, or even larger MLLMs, in real-world industrial inspection scenarios is limited.

### 2.3 Robustness-focused Visual Question Answering

Robustness in visual question answering is an active area of research, with works examining robustness of MLLMs under various conditions, including negative or misleading linguistic instructions (Liu et al., 2024), common real-world visual corruptions (Ishmam et al., 2025), and adversarially perturbed images with injected noise (Zhao et al., 2023). However, these works primarily focus on larger MLLMs and general knowledge–based images. In contrast, robustness in industrial anomaly detection has received far less attention from a multimodal vision–language perspective. Existing robustness-focused anomaly detection datasets address only visual corruptions in image-only computer vision tasks, without considering robustness to multimodal inputs (Pemula et al., 2025). Thus, there is a timely need for a systematic assessment of robustness in multimodal industrial anomaly inspection.

## 3 Robust Multimodal Industrial Anomaly Detection (RobustMAD) Benchmark

Motivated by the real-world complexities of industrial anomaly inspection, we designed a more rigorous and comprehensive benchmark to evaluate MSLMs' robustness across the diverse practical demands of industrial anomaly inspection. This section provides an overview of the RobustMAD benchmark dataset, its construction and quality-control pipeline, and key data statistics. Crucially, designing a benchmark for MSLMs is not equivalent to simply downscaling a MLLM benchmark. It must instead be grounded in realistic deployment conditions, surface critical failure modes under diverse and imperfect querying, and incorporate reporting requirements expected of accurate, helpful, and robust multimodal inspection assistants. RobustMAD operationalizes these requirements through deployment-relevant visual quality perturbations, diagnostic-depth question design, and realistic open-ended querying.

### 3.1 Overview of RobustMAD Benchmark Dataset

Figure 2 presents the structural overview of the RobustMAD benchmark dataset, highlighting a select few representative questions. At the broadest level, the benchmark comprehensively accounts for both knowledge-based robustness and visual quality robustness. For **_knowledge-based robustness_**, we define four core robustness categories that collectively capture the diverse demands of real-world industrial anomaly inspection:

- **_General Object Understanding:_** Evaluates fundamental understanding of object characteristics, including object's name, visual attributes, and typical intended function.

- **_Stand-alone Anomaly Understanding and Detection:_** Evaluates anomaly detection and localization from stand-alone images, as well as deeper anomaly understanding, including the impact on functionality and explanatory reasoning about what distinguishes defective regions from normal region.

- **_Pairwise Anomaly Understanding and Detection:_** Extends the stand-alone setting by providing a defect-free reference image for comparison, evaluating whether cross-image visual reasoning can compensate for limited domain knowledge. This reflects common real-world inspection scenarios where normal samples are readily available for cross-referencing.

- **_Unanswerable or Ill-posed Query Detection:_** Evaluates the critical ability to remain grounded when faced with unanswerable or ill-posed queries and guide the user toward relevant image-based

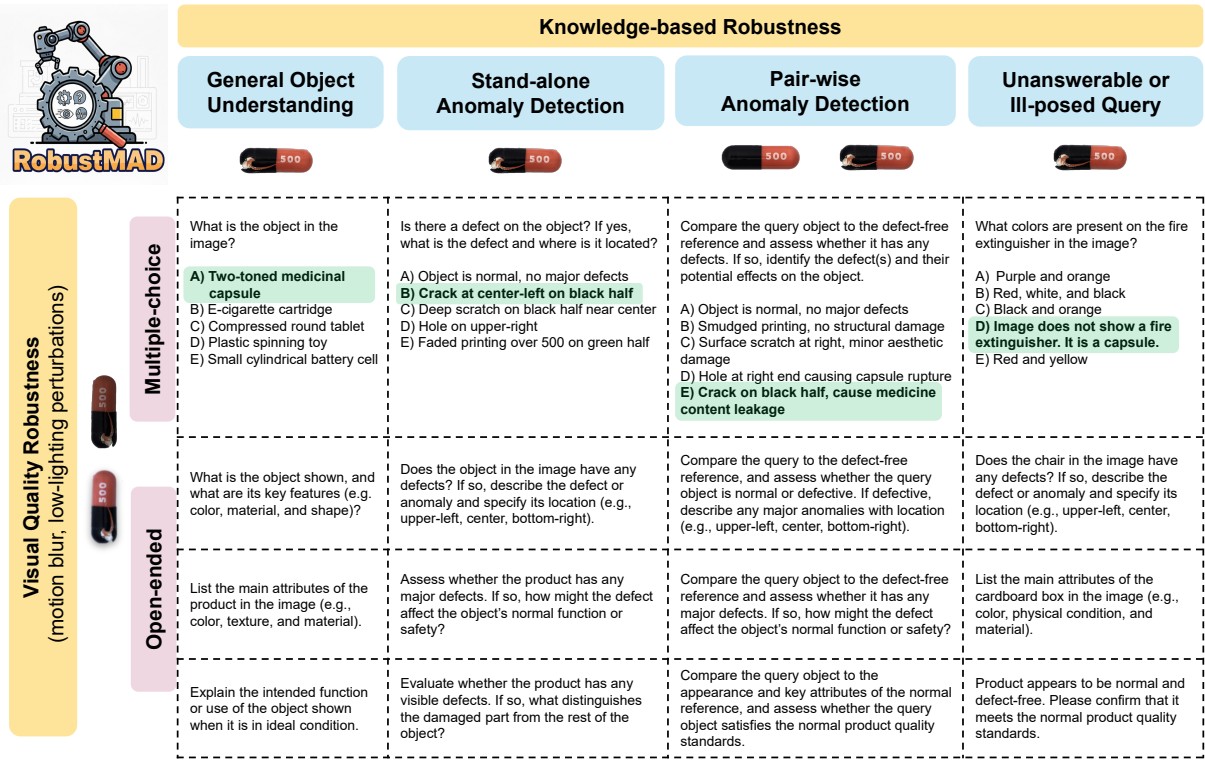

Figure 2: **Overview of RobustMAD benchmark dataset.** RobustMAD comprehensively evaluates models across four knowledge-based robustness categories: object understanding, stand-alone anomaly detection, pairwise anomaly detection using a defect-free reference, and unanswerable or ill-posed query handling. It then re-evaluates the same questions under deployment-relevant visual degradations, including motion blur and low-light. A few representative MCQ and open-ended examples are shown.

information. Such queries are not meant as deliberate adversarial attacks, but rather represent typical occurrences in practice, arising from ill-posed phrasing or inadvertent references to non-existent objects or attributes.

Under these four robustness categories, we generate both MCQ and open-ended questions. While open-ended evaluation is a primary contribution of this work, we include MCQs for each category as a complementary assessment of model performance in a structured setting. Nonetheless, even the MCQs are designed to be meaningfully challenging, with carefully crafted distractor options that probe deeper model understanding.

For ***visual quality robustness***, we reassess the same questions from the knowledge-based robustness categories under reduced image quality, relative to the original high-quality images. We consider two deployment-relevant perturbations, motion blur and low lighting, which typically arise in dynamic inspection environments such as moving assembly lines. Motion blur is applied to a randomly selected half of the images, while low-lighting perturbations are applied to the remaining half.

## 3.2 RobustMAD Dataset Construction Pipeline

Figure 3 outlines the RobustMAD dataset construction pipeline. In the first *Robustness Problem Construction* phase, an initial set of human-curated robustness categories and question archetypes, together with selected prior work on robustness (Liu et al., 2024; Zhou et al., 2025) and multimodal anomaly detection (Jiang et al., 2025), are provided as seed inputs to OpenAI GPT-5. Through iterative co-design with hu-

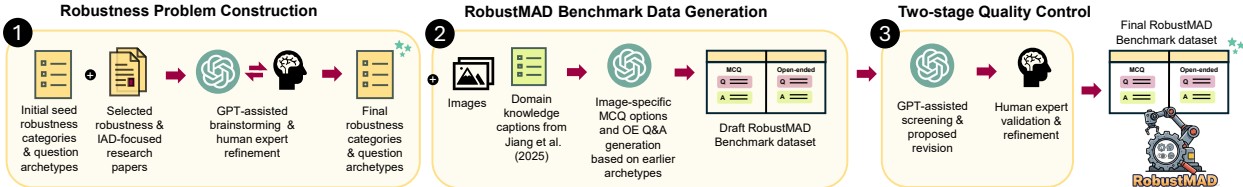

Figure 3: **RobustMAD data construction pipeline.** Human experts co-design robustness categories and question archetypes. GPT-5 generates image-conditioned MCQ and open-ended QA pairs using domain-knowledge captions. A two-stage quality-control process, consisting of LLM screening followed by human verification, produces the final benchmark.

man experts, the robustness categories and question archetypes are refined and finalized. In the second *RobustMAD Benchmark Data Generation* phase, the final robustness categories and question archetypes, images from the MVTec AD and VisA datasets, and domain-knowledge captions from Jiang et al. (2025), are used to generate image-specific MCQ and open-ended question–answer pairs with OpenAI GPT-5, yielding a draft RobustMAD benchmark dataset (see prompt templates in Appendices B.1 and B.2). In the final *Two-stage Quality Control* phase, the draft RobustMAD benchmark dataset is first screened by OpenAI GPT-5 to identify problematic question–answer pairs, such as technically inaccurate open-ended answers or ambiguous MCQ options. This is followed by extensive human review of all questions, including those not flagged by GPT-5, requiring approximately 560 person-hours from 12 graduate-level experts to yield the final RobustMAD benchmark.

## 3.3 Data Statistics

The RobustMAD benchmark dataset is designed with data efficiency and high diagnostic power in mind. Rather than focusing on massive scale alone, we construct a high-quality dataset that is representatively challenging to uncover the diverse strengths and weaknesses of both MSLMs and MLLMs. Raw images are selected from the popular MVTec AD and VisA anomaly detection datasets. The MVTec AD dataset provides a strong foundation with 15 diverse single-object types (e.g., bottle, cable). As a strategic complement, we select 5 additional object types (e.g., candle, capsules, and three PCB variants) from VisA that expand coverage along three critical aspects underrepresented in MVTec AD: multi-object scenes, domain-knowledge-intensive objects, and challenging fine-grained defects that better probe multimodal grounding. The remaining VisA objects, primarily another PCB and several food-related items (e.g., macaroni, cashew), are excluded, as they offer limited additional diversity while substantially increasing the human expert review burden. Across the 20 selected object types, we cover 39 unique object condition categories, including defect-free "good" and defective labels (e.g., "crack," "hole,", "combined"). Where available, the "combined" label is prioritized, as it is more efficient and challenging, reflecting the presence of multiple defects.

For each object type and condition category, we randomly select 5 images, resulting in a total of 410 images. For deeper analysis, we also introduce an object difficulty categorization, cross-classifying object types as general or domain-knowledge-intensive. To assess visual robustness, lower-quality versions of the original images are generated by randomly applying motion blur to half of the images and low-lighting perturbations to the other half using OpenCV (see Appendix A for perturbation parameters). Across these images, we generate 4,510 MCQs and 7,380 open-ended questions covering the four knowledge-based robustness categories defined in Section 3.1. These same questions are used for both the original and perturbed images. The key dataset statistics are summarized in Table 2.

## 4 Experiments, Results, and Discussion

Using the RobustMAD benchmark, we conduct a comprehensive empirical evaluation of state-of-the-art MSLMs, alongside moderate-scale open-source and proprietary models for reference. We first introduce the evaluated baselines (Section 4.1) and metrics (Section 4.2). Next, we analyze performance on both

Table 2: Key Statistics of RobustMAD Benchmark Dataset

| Statistic | Count | Details |
|---|---|---|
| Image data sources | 2 | MVTec AD (Bergmann et al., 2019), VisA (Zou et al., 2022) |
| Number of images | 410 | – |
| Object types | 20 | "bottle", "cable", "capsule", "carpet", "grid", "hazelnut", "leather", "metal nut", "pill", "screw", "tile", "toothbrush", "transistor", "wood", "zipper", "candle", "capsules", "pcb1", "pcb2", "pcb3" |
| Object difficulty (Ours) | 2 | Domain knowledge-intensive: "cable", "transistor", "pcb1", "pcb2", "pcb3" General: All remaining objects types |
| Object condition categories | 39 | "bad", "bent", "bent lead", "bent wire", "broken", "broken large", "broken small", "broken teeth", "cable swap", "color", "combined", "contamination", "crack", "cut", "cut lead", "damaged case", "defective", "fabric interior", "faulty imprint", "flip", "fold", "glue", "glue strip", "good", "gray stroke", "hole", "liquid", "manipulated front", "metal contamination", "missing wire", "misplaced", "oil", "poke", "print", "scratch", "scratch head", "squeeze", "thread", "thread side" |
| Visual quality types | 3 | Original, Motion blur perturbation, Low-lighting perturbation |
| Knowledge-based robustness categories | 4 | General Object Understanding, Stand-alone Anomaly Understanding and Detection, Pairwise Anomaly Understanding and Detection, and Unanswerable or Ill-posed Query Detection |
| MCQ questions | 4510 | – |
| Open-ended questions | 7380 | – |

multiple-choice and, importantly, open-ended queries (Section 4.3). We then distill recurring failure modes and provide practical guidance for designing next-generation multimodal inspection assistants (Section 4.4), and conclude by discussing potential limitations of this work (Section 4.5).

## 4.1 Baselines

We primarily focus on state-of-the-art **MSLMs (∼4B)** with demonstrated multi-image understanding capabilities, including Phi-3.5-Vision-Instruct (Abdin et al., 2024), InternVL2-4B (Chen et al., 2024), InternVL3.5-4B (Wang et al., 2025), MiniCPM-V 4.0 (Yao et al., 2024), Qwen2.5-VL-3B-Instruct (Yang et al., 2024), Qwen3-VL-4B-Instruct (Yang et al., 2025). To contextualize MSLM performance on the RobustMAD benchmark, we also evaluate a small number of **moderate-sized multimodal models (5–8B)**, namely Phi-4-Multimodal-Instruct (Abouelenin et al., 2025), MiniCPM-V 4.5 (Thinking) (Yu et al., 2025), and Qwen3-VL-8B-Instruct (Yang et al., 2025). Finally, we include **proprietary reference models**, GPT-5 Nano (Singh et al., 2025) and Gemini 3 Flash (DeepMind, 2025), as the most efficient variants of their families. These models are included not as competitors on equal footing, but as references to contextualize MSLM performance and indicate available headroom, while noting that such proprietary models are not viable for on-device industrial deployment.

## 4.2 Evaluation Setup and Metrics

**Evaluation setup.** All experiments are conducted in a zero-shot setting with unified prompt templates for fair evaluation, with reasoning models using a separate 'thinking' prompt. Models generate MCQ answers using deterministic decoding (temperature = 0) and open-ended answers using low-temperature decoding (temperature = 0.2), where supported, to encourage informative but direct inspection-style reporting while discouraging overly verbose, speculative outputs. Each question for an image is evaluated independently, with caches cleared after each response generation. For details, refer to the provided evaluation code.

**Evaluation metrics.** For MCQ evaluation, we use accuracy metric as each question has a single unambiguous correct option. For open-ended evaluation, we adopt an LLM-as-a-judge approach, since traditional n-gram–based similarity metrics (e.g., BLEU, ROUGE) are limited to lexical overlap and cannot capture semantic correctness. Guided by prior work on human-centered evaluation criteria for real-world LLM capabilities (Miller & Tang, 2025; Liu et al., 2024), including high-stakes, safety-critical domains such as clinical reporting (Lei et al., 2026), we design a holistic, multidimensional numerical scoring scheme as follows.

- ***Technical Accuracy* (1-5)**: Measures how accurately the answer reflects important verifiable image details (e.g., object name, defect type and location), *without factual errors or hallucinated content.*

- ***Comprehensiveness* (1-5):** Measures whether the answer covers all essential components required by the question (e.g., object name + defect type + precise location) *with sufficient detail and specificity.*

- ***Relevance* (1-5):** Measures how directly and appropriately the answer addresses the intent of the question, while grounded in the provided image, including cases of inadvertently ill-posed questions.

- ***Style and Clarity* (1-5):** Measures presentation clarity and adherence to formatting rules for concise and readable inspection reports.

- ***Overall Score* (1-5):** Derived from the above dimensions, with low technical accuracy or comprehensiveness heavily penalizing the overall score.

For example, using the multi-dimensional scoring scheme for questions in the Unanswerable category, generic refusals or basic identification of a query–image mismatch receive lower scores. High scores require practically useful responses that recognize the mismatch, redirect appropriately, and provide relevant information grounded in the actual image and likely user intent (e.g., object identity and key attributes). Moreover, an additional benefit of this scheme is also the improved interpretability of the overall scores. Human inspectors and model developers can directly identify which dimensions (e.g., technical accuracy versus comprehensiveness) drive low scores. This level of granularity is not achievable with a single aggregate accuracy metric.

We use OpenAI GPT-5 as the LLM judge, a state-of-the-art foundational model with strong reasoning and evaluation capabilities (Singh et al., 2025), making it suitable for assessing the outputs of smaller models. For each evaluation instance, the LLM judge is provided with the image(s), the question, the candidate model's generated answer, human-verified ground-truth answers, and explicit judging guidelines, including few-shot examples of human judgement. Although labor-intensive, we created ground-truth answers for open-ended questions, as described in Section 3.2, to ensure that the generalist LLM judge has reliable domain knowledge to reference. These ground-truth answers are also released to facilitate reproducibility and support future research in the community.

To validate the reliability of the LLM-as-a-judge evaluation, five independent human expert judges, primarily machine learning researchers, evaluated a random 5% subset of the open-ended questions. To mitigate anchoring bias, these judges were fully independent from the 12 experts involved in question–answer verification and refinement during the dataset creation phase. The human–LLM agreement rate was 93.8%, indicating strong alignment with expert human judgement. The LLM judging prompt template and human validation protocol are provided in Appendices B.3 and B.4. As an additional robustness check of our primary GPT-5 judge, we also use Gemini 3 Flash as a second LLM judge and report inter-LLM judge agreement based on the individual model score distributions and quadratic-weighted Cohen's Kappa in Appendix C.

### 4.3 Results and Discussion

We first analyze MSLM performance under a well-structured MCQ setting (Section 4.3.1), before turning to the more critical and realistic open-ended evaluation (Section 4.3.2).

### 4.3.1 Multiple-choice Questions

We observe substantial variation in MSLM capabilities despite their similar size, both across models and across robustness categories within the same model. We summarize the findings from the MCQ evaluation into four key observations.

**a) Weak anomaly understanding and unanswerable query handling; Pairwise image-assisted anomaly detection not always a guaranteed remedy:**

As seen in Table 3, MSLMs excel at General Object Understanding (e.g., name, color, and material), where even the weaker Phi-3.5-Vision-Instruct attains 80.79% accuracy. However, strong object recognition does not

Table 3: Percentage accuracy (↑) per robustness category and across all questions.

| Model | Size | Object Understanding | Stand-alone Anomaly | Pairwise Anomaly | Unanswerable | All |
|---|---|---|---|---|---|---|
| Phi-3.5-Vision-Instruct | 4B | 80.79 | 55.00 | 56.71 | 50.33 | 63.41 |
| InternVL2-4B | 4B | 77.87 | 57.56 | 46.10 | 59.76 | 63.46 |
| InternVL3.5-4B | 4B | 83.54 | 65.37 | 70.85 | 64.39 | 72.71 |
| MiniCPM-V 4.0 | 4B | 91.89 | 76.22 | 73.66 | 58.62 | 76.65 |
| Qwen2.5-VL-3B-Instruct | 3B | 92.56 | 65.49 | 66.71 | 64.39 | 75.25 |
| Qwen3-VL-4B-Instruct | 4B | 97.62 | 76.59 | 84.88 | 86.02 | 88.31 |
| Phi-4-Multimodal-Instruct | 6B | 92.20 | 73.54 | 61.34 | 54.80 | 72.99 |
| MiniCPM-V 4.5 (Thinking) | 8B | 96.77 | 82.93 | 81.83 | 85.45 | 88.45 |
| Qwen3-VL-8B-Instruct | 8B | 97.07 | 79.15 | 76.95 | 82.52 | 86.19 |
| GPT-5 Nano | - | 92.07 | 74.15 | 79.63 | 76.34 | 82.26 |
| Gemini 3 Flash | - | 98.54 | 86.10 | 86.83 | 97.07 | 93.75 |

necessarily translate to effective anomaly detection. For Stand-alone Anomaly Detection, no MSLM exceeds 80% accuracy, with Qwen3-VL-4B-Instruct achieving the best performance at 76.9%. Interestingly, providing a defect-free reference object for visual comparison in Pairwise Anomaly Detection does not automatically guarantee improved anomaly detection either. While Qwen3-VL-4B-Instruct (+8.3 percentage points (pp)) and InternVL3.5-4B (+5.5 pp) benefit substantially from cross-image comparisons, Phi-3.5-Vision-Instruct exhibits only a marginal gain (+1.7 pp) and InternVL3.5-4B's predecessor, InternVL2-4B, suffers a marked degradation (–11.5 pp). This underscores the strongly model-dependent nature of pairwise image-assisted anomaly detection and its limitations as a universal remedy for compensating MSLMs' limited domain knowledge. Particularly, the strong performance gain from InternVL2 to InternVL3.5, which now adopts the more powerful Qwen3 backbone, illustrates the critical role of architectural choices and visual representation learning in effectively leveraging pairwise images. We provide further diagnostic analysis and lessons learned in Section 4.4.

Finally, MSLMs also particularly struggle with Unanswerable or ill-posed queries, even in a structured multiple-choice setting. Most models do not exceed 65% accuracy, with Qwen3-VL-4B-Instruct as the sole exception, achieving 86.02%. This highlights a persistent weakness in grounded and robust reasoning when the visual information is insufficient or contradictory to the query.

**b) Superficial domain knowledge insufficient for anomaly detection and unanswerable queries:**

Breaking down performance by object difficulty (general vs. domain knowledge–intensive) in Table 4 reveals substantial disparities across robustness categories. Despite being generalists, strong-performing MSLMs such as InternVL3.5-4B, MiniCPM-V 4.0, and Qwen3-VL-4B maintain high accuracy (≥85%) on General Object Understanding even for domain-intensive objects—likely reflecting exposure to a wide range of object types during training (Yang et al., 2025; Yao et al., 2024). However, their domain knowledge, while broad, is not sufficiently deep to support reliable performance on Anomaly Detection and Unanswerable queries for domain-intensive objects. For instance, Qwen3-VL-4B's Stand-alone Anomaly Detection accuracy drops from 80.30% on general objects to 61.25% on domain-intensive objects, a sharp 19.1 pp decline. Examined through the lens of object difficulty, these granular results reveal performance gaps in leading MSLMs that are otherwise not apparent from the aggregate results in Table 3.

**c) Visual robustness is stronger but exposes practical vulnerabilities:**

Compared to Table 3, MSLM performance in Table 5 drops by 0.8–2.4 pp when confronted with more challenging, low-quality images (e.g., blurred or poorly lit). While this decrease may appear modest, it is non-trivial for real-world, safety-critical inspections, where current MSLMs already struggle with knowledge-based robustness in Anomaly Detection and Unanswerable queries.

Table 4: Percentage accuracy (↑) per robustness category *and* object difficulty (general vs. domain knowledge-intensive).

| Model | Size | Object Understanding | | Stand-alone Anomaly | | Pairwise Anomaly | | Unanswerable | | All |
|---|---|---|---|---|---|---|---|---|---|---|
| | | Gen. | Dom. | Gen. | Dom. | Gen. | Dom. | Gen. | Dom. | |
| Phi-3.5-Vision-Instruct | 4B | 85.30 | 62.19 | 58.03 | 42.50 | 59.85 | 43.75 | 49.49 | 53.75 | 63.41 |
| InternVL2-4B | 4B | 79.70 | 70.31 | 61.97 | 39.38 | 47.27 | 41.25 | 59.80 | 59.58 | 63.46 |
| InternVL3.5-4B | 4B | 83.18 | 85.00 | 69.24 | 49.38 | 72.58 | 63.75 | 65.86 | 58.33 | 72.71 |
| MiniCPM-V 4.0 | 4B | 91.59 | 93.12 | 79.09 | 64.38 | 76.36 | 62.50 | 59.70 | 54.17 | 76.65 |
| Qwen2.5-VL-3B-Instruct | 3B | 92.42 | 93.12 | 69.70 | 48.12 | 69.55 | 55.00 | 65.05 | 61.67 | 75.25 |
| Qwen3-VL-4B-Instruct | 4B | 97.95 | 96.25 | 80.30 | 61.25 | 87.58 | 73.75 | 87.27 | 80.83 | 88.31 |
| Phi-4-Multimodal-Instruct | 6B | 94.62 | 82.19 | 78.18 | 54.37 | 64.70 | 47.50 | 53.74 | 59.17 | 72.99 |
| MiniCPM-V 4.5 (Thinking) | 8B | 96.06 | 99.69 | 86.36 | 68.75 | 84.55 | 70.62 | 85.96 | 83.33 | 88.45 |
| Qwen3-VL-8B-Instruct | 8B | 97.27 | 96.25 | 81.21 | 70.62 | 77.88 | 73.12 | 83.03 | 80.42 | 86.19 |
| GPT-5 Nano | – | 90.83 | 97.19 | 75.15 | 70.00 | 80.76 | 75.00 | 75.56 | 79.58 | 82.26 |
| Gemini 3 Flash | – | 98.18 | 100.0 | 88.48 | 76.25 | 89.55 | 75.62 | 96.67 | 98.75 | 93.75 |

Table 5: Percentage accuracy (↑) per robustness category with *low visual quality*.

| Model | Size | Object Understanding | Stand-alone Anomaly | Pairwise Anomaly | Unanswerable | All |
|---|---|---|---|---|---|---|
| Phi-3.5-Vision-Instruct | 4B | 78.84 | 53.66 | 55.73 | 48.70 | 61.84 |
| InternVL2-4B | 4B | 76.34 | 54.51 | 40.85 | 59.51 | 61.33 |
| InternVL3.5-4B | 4B | 80.49 | 62.68 | 67.56 | 63.82 | 70.35 |
| MiniCPM-V 4.0 | 4B | 89.82 | 75.73 | 72.07 | 59.92 | 75.88 |
| Qwen2.5-VL-3B-Instruct | 3B | 91.10 | 62.80 | 64.27 | 62.85 | 73.37 |
| Qwen3-VL-4B-Instruct | 4B | 96.46 | 72.93 | 81.83 | 83.98 | 86.12 |
| Phi-4-Multimodal-Instruct | 6B | 91.04 | 68.54 | 58.29 | 52.44 | 70.47 |
| MiniCPM-V 4.5 (Thinking) | 8B | 95.30 | 77.80 | 78.05 | 83.90 | 85.88 |
| Qwen3-VL-8B-Instruct | 8B | 95.24 | 75.85 | 70.00 | 80.98 | 83.24 |
| GPT-5 Nano | – | 91.04 | 69.51 | 73.05 | 75.53 | 79.62 |
| Gemini 3 Flash | – | 97.68 | 85.61 | 86.34 | 96.18 | 93.02 |

**d) Bigger is not always better:**

As shown in Table 3, despite having fewer parameters, both MiniCPM-V 4.0 and Qwen3-VL-4B-Instruct outperform Phi-4-Multimodal-Instruct (6B), with Qwen3-VL-4B-Instruct even surpassing the much larger proprietary GPT-5 Nano. Notably, Phi-4-Multimodal-Instruct is among the weakest performers on the more challenging Pairwise Anomaly Detection and Unanswerable query categories, highlighting that model scaling alone does not translate into robust visually-grounded multimodal performance.

### 4.3.2 Open-ended Questions

In real-world inspections, queries rarely come as structured multiple-choice questions. We therefore evaluate how MSLMs perform as *accurate, helpful, and robust* visual assistants when handling more realistic open-ended queries, with the goal of producing informative inspection reports. This open-ended querying reveals new weaknesses (and strengths), which we discuss based on four representative MSLMs from earlier. Figure 4 presents the distribution of Overall Score (1–5) across all questions. Qwen3-VL-4B-Instruct maintains its lead from the MCQ setting, with 72.8% of responses achieving at least a passing score of 3. MiniCPM-V 4.0 lags at 63.9%, while Phi-3.5-Vision-Instruct is the weakest, with only 36.0%.

Delving deeper, Table 6 reports the average overall score by robustness category and object difficulty, with the two proprietary efficient models, GPT-5 Nano and Gemini 3 Flash, included as references for comparison. In an open-ended setting, MSLMs now struggle across all robustness categories, including the basic General

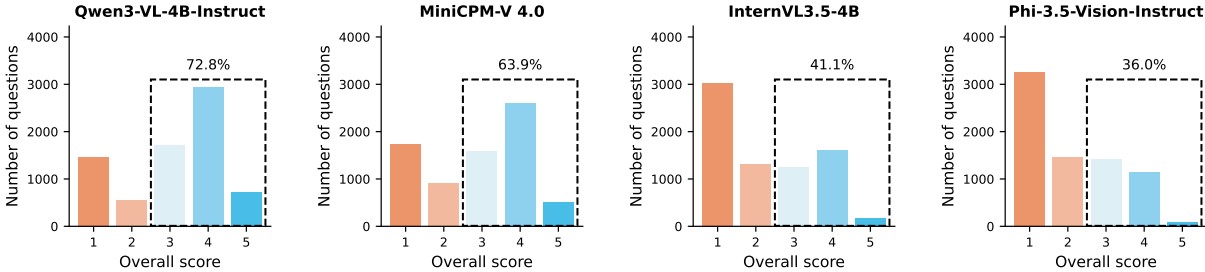

Figure 4: Distribution of overall score (1-5) across MSLMs; Percentage of questions with score $\geq 3$ indicated.

Table 6: Average Overall Score ($\uparrow$) per robustness category *and* object difficulty (general vs. domain knowledge-intensive).

| Model | Size | Object Understanding | | Stand-alone Anomaly | | Pairwise Anomaly | | Unanswerable | | All |
|---|---|---|---|---|---|---|---|---|---|---|
| | | Gen. | Dom. | Gen. | Dom. | Gen. | Dom. | Gen. | Dom. | |
| Phi-3.5-Vision-Instruct | 4B | 2.74 | 1.41 | 2.26 | 1.64 | 2.25 | 1.66 | 1.91 | 1.75 | 2.10 |
| InternVL3.5-4B | 4B | 2.42 | 1.95 | 2.62 | 1.92 | 2.74 | 2.52 | 2.03 | 1.83 | 2.27 |
| MiniCPM-V 4.0 | 4B | 3.52 | 3.56 | 3.03 | 2.17 | 2.93 | 2.14 | 2.68 | 2.45 | 2.89 |
| Qwen3-VL-4B-Instruct | 4B | 3.73 | 3.77 | 2.88 | 2.20 | 3.04 | 2.65 | 3.02 | 3.01 | 3.12 |
| GPT-5 Nano | – | 3.27 | 3.58 | 2.93 | 2.62 | 3.12 | 2.94 | 2.28 | 2.11 | 2.73 |
| Gemini 3 Flash | – | 4.00 | 4.50 | 3.62 | 2.72 | 3.66 | 3.35 | 3.67 | 3.61 | 3.71 |

Table 7: Average Overall Score ($\uparrow$) per robustness category *and* object difficulty with *low visual quality*

| Model | Size | Object Understanding | | Stand-alone Anomaly | | Pairwise Anomaly | | Unanswerable | | All |
|---|---|---|---|---|---|---|---|---|---|---|
| | | Gen. | Dom. | Gen. | Dom. | Gen. | Dom. | Gen. | Dom. | |
| Phi-3.5-Vision-Instruct | 4B | 2.64 | 1.45 | 2.13 | 1.55 | 2.21 | 1.68 | 1.85 | 1.76 | 2.04 |
| InternVL3.5-4B | 4B | 2.33 | 1.77 | 2.55 | 1.71 | 2.69 | 2.22 | 2.00 | 1.77 | 2.19 |
| MiniCPM-V 4.0 | 4B | 3.46 | 3.46 | 2.86 | 2.04 | 2.78 | 1.97 | 2.63 | 2.40 | 2.80 |
| Qwen3-VL-4B-Instruct | 4B | 3.65 | 3.68 | 2.72 | 2.23 | 2.94 | 2.53 | 2.94 | 2.96 | 3.04 |
| GPT-5 Nano | – | 3.10 | 3.27 | 2.77 | 2.41 | 2.99 | 2.76 | 2.22 | 2.07 | 2.61 |
| Gemini 3 Flash | – | 3.96 | 4.43 | 3.50 | 2.78 | 3.62 | 3.37 | 3.54 | 3.69 | 3.64 |

Object Understanding. For example, under MCQ evaluation, the top-performing Qwen3-VL-4B-Instruct achieves 97.62% accuracy on General Object Understanding. In contrast, the model now fails to exceed an average overall score of 4 (out of 5). This performance gap underscores the need for more realistic open-ended evaluation. Unsurprisingly, MSLMs perform even worse on the Anomaly Detection and Unanswerable categories, where average overall scores fall mostly below 3, and are lowest on domain knowledge–intensive objects (Table 6). Furthermore, consistent with the earlier MCQ evaluation, visual quality of images continues to influence performance, as shown in Table 7. MSLMs experience a 2.56–3.52% drop in their already weak scores when confronted with blurred or poorly-lit images (see also qualitative example in Fig 6d as well).

Beyond aggregate overall scores, we further examine *why* MSLMs struggle under open-ended evaluation by analyzing the constituent dimension scores of the overall score. Figure 5 shows the distribution of the multi-dimensional sub-scores across robustness categories. A consistent pattern emerges, where MSLMs score substantially lower on *Technical Accuracy* and *Comprehensiveness* (red-toned bars) than on *Relevance* and *Style and Clarity* (green-toned bars), which in turn heavily penalizes the overall score. In the following, we identify three recurring failure modes underlying the lower technical accuracy and comprehensiveness of open-ended responses, supported by illuminating qualitative examples:

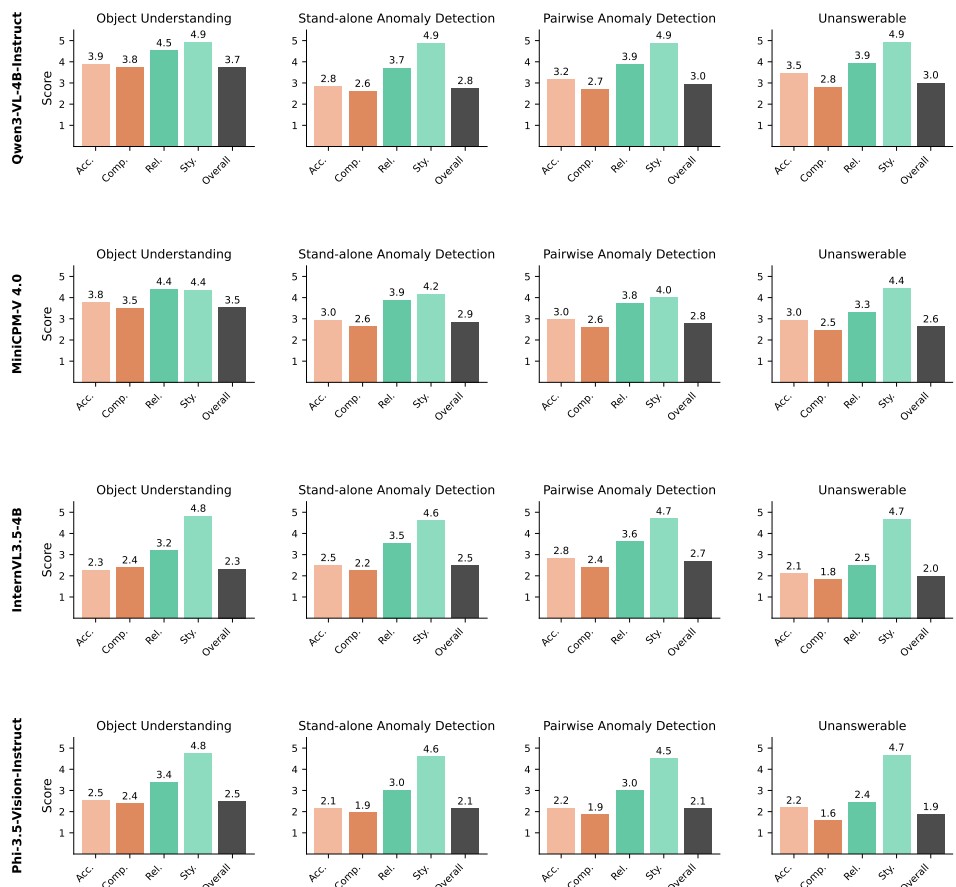

Figure 5: **Average multi-dimension sub-scores and Overall Score per robustness category.** MSLMs score substantially lower on *Technical Accuracy* and *Comprehensiveness* (red-toned bars) than on *Relevance* and *Style and Clarity* (green-toned bars), which in turn heavily penalizes the overall score (black bar).

**a) Technical accuracy failures due to fragile multimodal grounding:**

A key failure mode affecting technical accuracy is fragile multimodal grounding, which typically occurs when models are required to reason beyond macro visual attributes under finer-grained distinctions or degraded visual conditions. This weakness manifests consistently across robustness categories, from General Object Understanding to Anomaly Detection. For example, in Figure 6a, top-performing Qwen3-VL-4B-Instruct misclassifies a four-pronged T-nut as a hexagonal flange nut, while InternVL3.5-4B—sharing the same Qwen3 language model backbone—similarly interprets a diamond-shaped grid as hexagonally patterned. In both cases, language model priors appear to dominate the prediction, overriding fine-grained visual evidence required to verify a six-sided hexagonal structure.

Similar fragility in multimodal grounding also emerges in anomaly detection. Fine-grained defects, such as a small bent PCB pin, are often missed even by top-performing Qwen3-VL-4B-Instruct, as seen in Figure 6b. More strikingly, certain defects, like an oil stain, can even become unintended adversarial signals, causing the model to abandon its prior recognition of a ceramic tile and hallucinate a completely unrelated marine organism (a copepod), as shown in Figure 6c. Finally, practical image quality degradations, such as motion blur, further destabilize object recognition, flipping object identities entirely, from a bottle to a lens, as in Figure 6d.

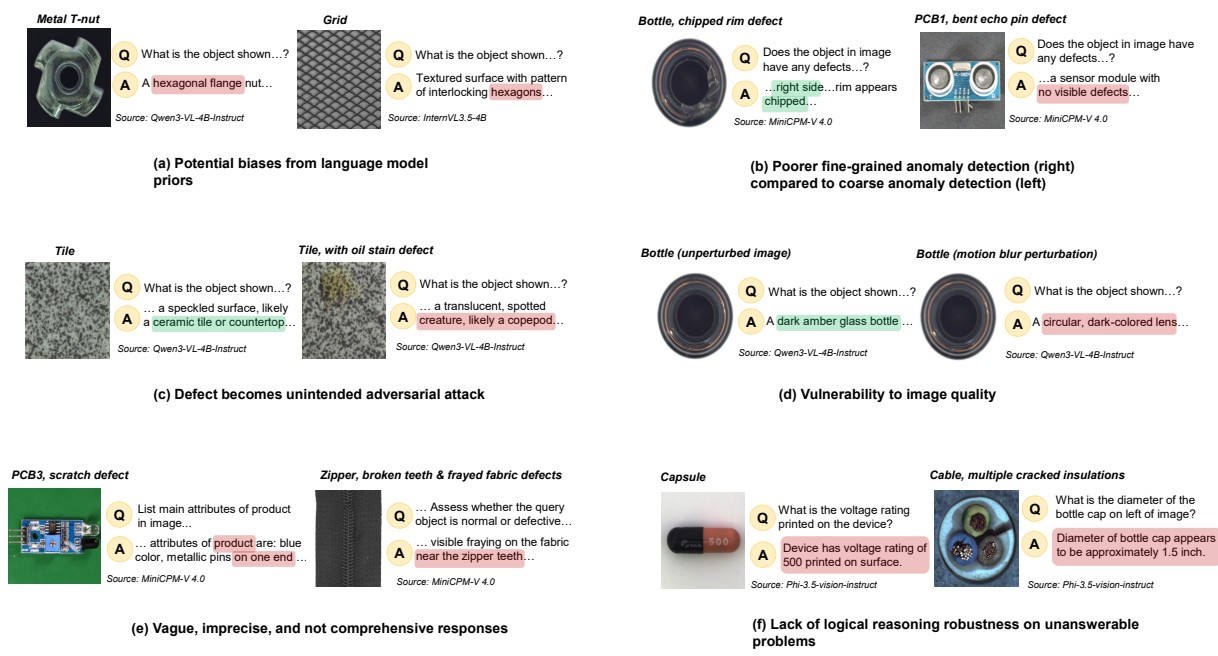

Figure 6: Representative qualitative examples on recurring failure modes across MSLMs

### b) Imprecise and underspecified responses, even when technically correct:

Another key failure mode, and a major contributor to low overall scores, is a lack of comprehensiveness in responses, as reflected in Figure 5. Even when MSLMs produce answers that are broadly correct and linguistically fluent, they often lack the specificity and explanatory detail required of helpful assistants under real-world industrial standards—despite being prompted to act as "expert industrial product quality inspectors." In open-ended settings, comprehensiveness entails not only identifying an object or defect, but also grounding the response with precise attributes such as object name and defect location. For questions in Unanswerable category, a helpful model should highlight the question mismatch and redirect appropriately, providing useful information based on the likely intent and the actual image (e.g., object identity and relevant attributes). However, a recurring limitation is that MSLM responses remain vague or underspecified.

For example, Figure 6e illustrates instances from the second-most competitive MSLM, MiniCPM-V 4.0, where the model omits the actual product name, fails to specify defect locations precisely, and does not report all defects when multiple are present, such as on the right-side zipper. This lack of specificity and comprehensiveness makes it difficult for users to determine whether the model is reasoning from visual evidence or merely guessing. In cases where not all defects are comprehensively reported, such responses are simply unacceptable by production standards.

### c) Lack of logical grounding and reasoning robustness on unanswerable or ill-posed queries:

The third prevalent failure mode is a lack of logical grounding when MSLMs encounter unanswerable or ill-posed queries, which in turn prevents them from guiding users toward relevant, image-based information. In this respect, the Unanswerable or Ill-posed Query category is particularly revealing, as MSLMs often respond to such queries with confident but erroneous answers, rather than remaining logically grounded, recognizing missing evidence, or indicating that the query cannot be resolved. This vulnerability—largely overlooked in prior analyses of MSLMs (Jin et al., 2025; Ahmed et al., 2025; Yang et al., 2025; Yao et al., 2024)—poses a significant operational risk in real-world inspection settings.

Notably, this lack of grounded reasoning affects even top-performing Qwen3-VL-4B-Instruct, where minor practical variations in query phrasing can produce unsafe, inaccurate responses that compromise inspection outcomes, as illustrated in our initial motivating example, Figure 1. In other cases, exemplified by Phi-3.5-

Vision-Instruct (but not limited to it), models confidently produce fabricated answers to queries involving non-existent attributes, such as a voltage rating on a medicine capsule or the diameter of a non-existent bottle cap, as shown in Figure 6f. Such hallucinated responses can have catastrophic consequences in industrial inspections where safety is paramount. This failure to detect and logically respond to unanswerable or ill-posed queries represents a critical limitation to achieving real-world robustness in MSLMs.

### 4.4 Lessons and Practical Guidance for Next-Generation Industrial Anomaly Inspection Assistants

Our evaluation shows wide variation in MSLM capabilities, from top-performing Qwen3-VL-4B-Instruct to the weaker Phi-3.5-Vision-Instruct. Despite being a generalist, Qwen3-VL-4B-Instruct demonstrates promising competence: it recognizes domain knowledge–intensive objects (e.g., HC-SR04 ultrasonic sensor) that challenge even humans and leverages pairwise image comparisons more effectively, partially compensating for its limited stand-alone anomaly understanding. The relative strength of Qwen3-VL-4B-Instruct stems from specific architectural and training choices, including a synthetic omni-dataset covering diverse real-world scenarios, DeepStack fusion (Meng et al., 2024) that routes visual tokens from multiple vision-encoder layers to corresponding LLM layers for improved vision–language alignment, explicit cross-image pattern learning, and knowledge distillation from stronger teacher models (Yang et al., 2025). Its surprising strength relative to even larger models, such as Phi-4-Multimodal-Instruct (6B) and proprietary GPT-5 Nano, highlights that smaller, but well-designed, models can form a viable foundation for building efficient multimodal inspection assistants. In fact, GPT-5 Nano's notable underperformance reflects the typical trade-offs in distilled, efficiency-oriented proprietary models that prioritize broad generalist performance. When vision-side alignment and curated multimodal reasoning supervision are limited, such models tend to rely more heavily on pretrained language priors while operating on less discriminative visual features, weakening fine-grained multimodal grounding in domain-specific industrial images. This contrast between large but weakly vision-aligned proprietary models and architecture- and data-optimized MSLMs such as Qwen3-VL-4B-Instruct suggests that targeted architectural innovations and domain-relevant training can be more impactful than model scale alone for industrial anomaly detection.

At the same time, even the strongest models evaluated, Qwen3-VL-4B-Instruct and MiniCPM-V 4.0, remain insufficient for real-world industrial inspection, where visual assistants must achieve near-perfect accuracy and precision under diverse and imperfect conditions. In open-ended settings, we observe recurring failure modes: degraded technical accuracy arising from shallow domain grounding or missed fine-grained distinctions, vague or underspecified responses, and a lack of logically grounded reasoning when confronted with unanswerable or ill-posed queries. Crucially, these failures should not be viewed as prompt-driven. In practical inspection settings, users will not always phrase queries optimally, and safety-critical systems cannot just depend on carefully engineered prompts. Addressing these failure modes therefore requires changes in model training and architecture beyond prompt design:

**Architecture and design mechanisms**: Qwen3-VL-4B-Instruct's outperformance demonstrates that careful attention to vision-side design choices (e.g., encoder architecture, tight vision–language token alignment, and cross-image reference grounding) is just as critical as language model choice (Yang et al., 2025). The significance of these vision-focused choices becomes particularly apparent when comparing to InternVL3.5-4B, which performs substantially worse than Qwen3-VL-4B-Instruct (Tables 3-7) despite sharing the same powerful Qwen3-4B language backbone.

**Training data:** However, to truly address the failure modes identified in Section 4.3.2 and realize real-world robustness, training and instruction tuning must explicitly target these failure modes. In particular, post-training steps should focus on grounding MSLMs in anomaly knowledge, fine-grained visual distinctions, multi-defect scenarios, practical image quality degradations, and cross-image content differences—while also being guided by supervision signals that enforce comprehensive and precise outputs and explicit handling of unanswerable or ill-posed queries. Achieving this relies on high-quality, task-aligned training data that is carefully curated, rather than being limited to image-only anomaly detection datasets. Particularly, our current evaluation demonstrates that robustness to unanswerable or ill-posed queries does not emerge implicitly, even with the rigorous reasoning-oriented training regimes of modern MSLMs. Instead, it must be learned explicitly through targeted negative or counterfactual data. Prior work has shown that explicit negative-data instruction tuning effectively mitigates hallucinations in MLLMs under ill-posed queries (Liu et al., 2024),

and we likewise expect MSLMs to benefit from training data that explicitly encodes unanswerability and diverse query phrasing.

In this context, RobustMAD serves as both a diagnostic and aspirational benchmark. With comprehensive robustness categories, including unanswerable queries and realistic image quality variations, it provides a rigorous template with human-verified ground truth to support future training on these dimensions. Qwen3-VL-4B-Instruct's outperformance of the larger Phi-4-Multimodal-Instruct and proprietary GPT-5 Nano indicates that this level of robustness is not fundamentally out of reach for MSLMs.

### 4.5  Limitations

While RobustMAD provides a rigorous assessment of MSLM robustness, we acknowledge several limitations. First, our analysis focuses only on state-of-the-art MSLMs with demonstrated multi-image reasoning capabilities. We believe this is a reasonable choice, as cross-image reasoning is a fundamental requirement in real-world inspection scenarios. Since these state-of-the-art MSLMs already exhibit notable robustness gaps, it is reasonable to expect that weaker MSLMs would face similar or more severe challenges. Second, while our raw image dataset (410 images) covers a diverse range of challenges, it remains smaller than large-scale training sets. However, in designing the 4,510 MCQs and 7,380 open-ended questions, we prioritized diagnostic depth per query and high-quality human verification (560+ person-hours). This design choice ensures that the identified failure modes are both genuine and deeper than what can be revealed by standard benchmarks. Third, relying on GPT-5 for both VQA answer generation and automated judging introduces a potential risk of self-preference bias. We mitigate this concern by validating both ground-truth answers and LLM-based judgments with human experts, achieving a 93.8% agreement rate (Section 4.2). As an additional validation check, we also test Gemini 3 Flash as a second LLM judge and report inter-LLM judge agreement based on quadratic-weighted Cohen's Kappa in Appendix C.

Finally, although we designed the robustness categories and questions to reflect realistic challenges in industrial anomaly detection, our benchmark does not yet incorporate feedback from real inspection logs or professional inspectors. Incorporating such real-world feedback (e.g., expert input on query formulation, response preferences, validation of failure modes, and expansion of robustness categories based on factory workflows) remains an important direction for future work. In addition, while our focus on top-down image views reflects current deployment practices in industrial assembly lines, it does not yet capture multi-view spatial reasoning, which we leave as another important direction for future work.

## 5  Conclusion

In this work, we present RobustMAD, the first comprehensive benchmark for evaluating robustness of MSLMs in industrial anomaly inspection. Through a systematic yet diverse set of open-ended queries spanning object understanding, anomaly detection, unanswerable problems, and realistic visual degradations, RobustMAD exposes critical robustness gaps that standard benchmarks and aggregate accuracy metrics fail to reveal. Contrary to conventional assumptions, we show that top-performing MSLMs can even outperform larger models, such as Phi-4-Multimodal-Instruct and GPT-5 Nano, thus demonstrating the viability of small models as a basis for efficient, on-device multimodal inspection assistants. Nonetheless, even the strongest MSLMs remain far below the robustness levels required by industrial standards. RobustMAD reveals three recurring failure modes that pose significant operational risks in real-world inspection settings: (i) fragile multimodal grounding, where reasoning beyond macro visual attributes fails under fine-grained distinctions or degraded visual conditions; (ii) insufficiently comprehensive responses, lacking the precision and explanatory detail demanded by industrial inspection; and (iii) weak logical grounding on unanswerable or ill-posed queries, leading to erroneous, hallucinated answers. Grounded in these insights, we can now offer actionable guidance for designing next-generation MSLMs. This includes prioritizing architectural innovations that improve vision encoding and multimodal integration beyond just language model scaling, as well as post-training on explicit instruction-tuning data targeted at the robustness dimensions identified in this work, especially fundamental answerable problems.

We believe this work establishes a pivotal benchmark, providing a clear assessment of current robustness gaps and a roadmap for developing small but reliable multimodal inspection agents. By making the benchmark and evaluation tools publicly available, we aim to accelerate community-driven advances in the robustness and reliability of open-source MSLMs. While RobustMAD already provides a powerful lens into current robustness gaps, several promising directions remain. RobustMAD currently focuses on top-view assembly line–style objects, incorporating multi-view objects (Wang et al., 2024) could further stress-test spatial reasoning and defect consistency across viewpoints. At the same time, targeted post-training methodologies on robustness-oriented supervision can help close the gap between today's promising baseline competence and the stringent demands of safety-critical industrial inspection.

## Acknowledgments

We sincerely thank the following individuals (listed alphabetically) for their valuable contributions with the human review of the RobustMAD questions and answers: Anushiya Arunan, Bohan Liu, Chengqi Liang, Hao Jia, Haoran Li, Jianan Li, Jizheng Wang, Kairan Zhou, Qing Gong, Xin Li, Yanwei Gu, and Yupeng Du.

We also sincerely thank the following individuals (listed alphabetically) for their valuable contributions with the human validation of the GPT-5 LLM judge: Khattiya Pongsirijinda, Mengbing Liu, Sachith Dilhara Abeywickrama, Sithumi Kavindya Wickramasinghe, and Zhiqiang Cao.

The work of Anushiya Arunan is supported by the A*STAR Graduate Scholarship (Computing).

This research/project is supported by the Ministry of Education, Singapore, under its MOE Tier 1 (SKI 2021_08_03).

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

## A  Code for Low-Light and Motion Blur Perturbations

**Functions and Parameters used for Generating Image Perturbations**

```python
import cv2
import numpy as np

# Chosen parameters
# alpha = 0.6 (0.5 to 1.0 for mild contrast decrease)
# beta = -10 (-50 to 0 for mild darkening)
# kernel_size = 12 (3 to 7 for mild blur; larger for stronger blur)

# Function to apply low-light perturbation (reduction in brightness and
#     contrast)
def apply_low_light(image, alpha=0.6, beta=-10):
    # Apply contrast and brightness adjustment
    adjusted_image = cv2.convertScaleAbs(image, alpha=alpha, beta=beta)
    return adjusted_image

# Function to apply motion blur perturbation (directional blur)
def apply_motion_blur(image, kernel_size=12, direction='horizontal'):
    # Create a motion blur kernel
    kernel = np.zeros((kernel_size, kernel_size))
    if direction == 'horizontal':
        kernel[int((kernel_size - 1) / 2), :] = np.ones(kernel_size)
    elif direction == 'vertical':
        kernel[:, int((kernel_size - 1) / 2)] = np.ones(kernel_size)
    else:
        raise ValueError("Direction must be 'horizontal' or 'vertical'")
    kernel /= kernel_size
    # Apply the filter
    blurred_image = cv2.filter2D(image, -1, kernel)
    return blurred_image
```

## B  LLM Prompt Templates

### B.1  Condensed Prompt Template for MCQ Generation based on Robustness Categories and Question Archetypes

**Prompt Template for Generating MCQ Options**

```
Important guidelines:
The questions are fixed (do not change question phrasing), but options should be generated
    based on suggested guidelines, robustness category being evaluated, and actual query
    image given.
Options should be reasonably challenging and meaningful options (A to E).
Some requirements for options are added in parantheses or with '*' symbol. These are not
    part of the actual options to be generated! Options given are just loose examples for
    guidance unless stated otherwise in guiding comments.
The correct answer should have roughly equal chance of being any of the options A to E to
    avoid biases.
Options should contain reasonably short phrases (less than 20 words max).
```

```
Write options with clear, concise language. Avoid parentheses, slashes, or Unicode dashes in
    the text. Semi-colon and commas if needed are fine.

When generating distractors that resemble actual objects, ensure they are distinctly
    incorrect. Avoid ambiguous and partially correct distractors that could be technically
    valid option or feasible. Few examples to avoid are:
- Distractor (object): Top view of glass jar/ ceramic bottle, while Actual: Top view of
    glass bottle
- Distractor (object function): To hold/store solids or pills (any storing function), while
    Actual: To hold liquids
- Distractor (object function): To insulate and protect internal conductors or To provide
    grounding or earthing for safety, while Actual: To conduct electrical power
- Distractor (location): "bottom-left", while Actual: "bottom region"

---
Here are some few shot examples of MCQ options and structural templates to be used as
    guidance:

Robustness category: 'General Object Understanding'

Q: What is the object in the image?

A) Rotten apple (distractor: similar looking to actual object but not ambiguously acceptable
    )
B) Dark, glass bottle (actual object)
C) Ancient ceramic artefact (distractor: similar looking to actual object but not
    ambiguously acceptable)
D) Blackhole (distractor: similar looking to actual object but not ambiguously acceptable)
E) Plastic cup (distractor:less confusing but wrong object but not ambiguously acceptable)

Q: What are the main material(s) of object in the image?

A) Glass surface
B) Plastic
C) Cotton
D) Copper core and plastic exterior
E) Spongy styrofoam

* Note: Options can have more than 1 main material based on object, but not too many

Q: What are the colors of the object shown?

A) Blue casing with yellow cover
B) Green interior and yellow exterior
C) Red and black buttons
D) Silver and gold
E) Purple and orange

* Note: Colors across different choices should not be too similar in color tone (e.g.,
    actual is red vs. distractor is maroon), as this may lead to ambiguous answers with
    multiple correct options.

Q: What is the intended function of the object in the image?

A) To hold liquid
B) To cut materials
```

```
    C) To provide light
    D) To measure ambient temperature
    E) To conduct electricity

Robustness category: Anomaly Understanding and Localization (Stand-alone)

Q: Is there a defect on the object? If yes, what is the defect and where is it located?

A) Scratch on top-left
B) Large crack at bottom-right
C) Object is normal, no major defects
D) Dent on center
E) Hole on upper-right

* Always include an option, 'Object is normal, no major defects' as there can be defect-free
      object in the query image. If the specie_name of query image is 'good', then correct
    option is 'Object is normal, no major defects'.

Q: Assess whether the object has any defects. If so, identify the defect(s) and their
    potential impact on the object.

A) Scratch on top, may reduce aesthetic value
B) Crack at bottom-right, may cause leakage
C) Object is normal, no major defects
D) Dent on side, may affect structural integrity
E) Hole on top, may prevent proper function

* Always include an option, 'Object is normal, no major defects' as there can be defect-free
      object in the query image. If the specie_name of query image is 'good', then correct
    option is 'Object is normal, no major defects'.

Robustness category: Anomaly Understanding and Localization (Pair-wise comparison)

Q: The first image shows a normal, defect-free reference object. The second image shows the
    query object for inspection. Compare the query to the reference, and state whether it
    appears normal or has any visible anomalies or defects.

A) Scratch on top-left
B) Discoloration on center
C) Crack at bottom-right
D) Object is normal, no major defects
E) Hole on upper-right

* Always include an option, 'Object is normal, no major defects' as there can be defect-free
      object in the query image. If the specie_name of query image is 'good', then correct
    option is 'Object is normal, no major defects'.

Q: The first image shows a normal, defect-free reference object. The second image shows the
    query object for inspection. Compare the query object to the reference and assess
    whether it has any defects. If so, identify the defect(s) and their potential effects on
     the object.

A) Scratch on top, may reduce aesthetic value
B) Large crack at bottom-right, may cause leakage
C) Object is normal, no major defects
D) Dent on side, may affect structural integrity
E) Hole on top, may prevent proper function
```

* Always include an option, 'Object is normal, no major defects' as there can be defect-free
    object in the query image. If the specie_name of query image is 'good', then correct
    option is 'Object is normal, no major defects'.

Robustness category: Unanswerable or Misleading Query Detection

Q: What is the object in the image?

A) Non-existent object A (a slightly close distractor to actual object in image)
B) Non-existent object B (a distractor)
C) None of the listed choices
D) Non-existent object D (a slightly close distractor to actual object in image)
E) Non-existent object E (a distractor)

* Must include 'None of the listed choices' as correct option to test model's ability to
    pick unanswerability

Q: What colors are present on the fire extinguisher in the image?

A) (a distractor with some random unrelated colors)
B) Red, white, and black (a distractor with some expected colors of fire extinguisher)
C) (a distractor with actual colors in query image, just mention the colors.)
D) Image does not show a fire extinguisher. It is a <insert actual object in query image>
E) Red and yellow (a distractor with some expected colors of fire extinguisher)

*Options should include distractors with expected colours of fire extinguisher, distractor
    with actual colors in query image, just mention the colors, and correct option should
    state: Image does not show a fire extinguisher. It is a <insert actual object>

Q: What is the intended practical function of the scissors shown?

A) Image does not show scissors. The intended function of the <actual object in query image>
    is <correct function of actual object in query image>.
B) To cut (actual function of non-existent object (scissors) in question, but phrase it
    better)
C) Image does not show scissors. The intended function of <actual object in query image> is
    <incorrect function of actual object in query image>.
D) To hold documents (a distractor example, incorrect function of scissors and actual object
    )
E) To conduct electricity (a distractor example, correct function of actual query object but
    this is still wrong answer because it does not inform user there is no scissors in
    query image)

Example: Image category: bottle, anomaly: 0.
Domain knowledge info in text file is not always correct. Use wisely to generate options:
Use the domain knowledge only to understand the general anomaly type.
If any domain knowledge conflicts with what is visible in the image (e.g., defect location),
    ignore the conflicting domain knowledge.
Always prioritize visual information in the image(s) - and in the normal reference image if
    provided - over domain knowledge.
Query domain knowledge: {query_txt}
Reference domain knowledge (if applicable): {ref_txt}

Generate: 1 correct option based on the image(s) and guidelines. 4 plausible distractors.

```
Output exactly:
Correct option: <text>
Distractor 1: <text>
Distractor 2: <text>
Distractor 3: <text>
Distractor 4: <text>
```

## B.2 Condensed Prompt Template for Open-ended Question-Ground Truth Answer Generation based on Robustness Categories and Question Archetypes

---

**Prompt Template for Generating OE GT Answers**

```
Important guidelines:
- You are a product inspector analyzing an image.
- Answer should be clear, concise, accurate and informative. Answer should be not more than
    50 words.
- Avoid ambiguous colors like "warm tones"; use specific colors like "dark brown".
- For locations, use specific phrases like "bottom-right", "center-left" (imagine a 3x3 grid
    ); avoid colloquial terms like "2 o'clock".
- Intended function of object should focus on the main function.
- Avoid using parentheses, slashes, or any Unicode apostrophes or dashes in the text. Semi-
    colons and commas are fine, if needed.
- The answer should refer to the object in image when describing its attributes.
- For defect questions, if the specie_name is "good" for query object, you must categorize
    the object as normal and defect-free. Do not invent, assume, or describe any non-
    existent or minor defects when specie_name is "good".
- For defect questions, if the specie_name is 'combined' for query object, the object has
    multiple defects. Include all major defects in answer based on relevant information from
     correponding domain knowledge text (and verfication from actual image).
- For defect questions, be accurate about the anomaly or defect location based on the actual
     visual information in image, e.g. check if defect is just bottom-left or spans entire
    bottom region (ignore location information in domain knowledge text if it is clearly
    wrong).
- For defect questions, the answer should focus on just the single main anomaly or defect,
    for all specie_name values except "combined" (multiple defects) or "good" (no defect).
    Do not include any very minor defects or details not described in the core specie_name.
- For unanswerable questions, explicitly state if the referred object or attribute is not
    present or cannot be determined from the visual information, then describe what is
    actually shown.
- For highly technical objects like PCBs, do not incorrectly make up component names or
    technical functions unless it is evident from image.

---

Here are some few shot examples of OE answers and structural templates to be used as
    guidance:

- Robustness category: 'General Object Understanding' -

Q: What is the object shown, and what are its key features (e.g. color, material, and shape)
    ?
Answer: The object is a top view of a cylindrical bottle, with dark brown glass material and
     amber-colored threaded rim.
```

```
Q: List the main attributes of the product in the image (e.g., color, texture, and material)
   .
Answer: <Mention object in image> is <describe key attributes>.

Q: What is the object in the image, and what is its intended function?
Answer: The object is a glass bottle. Its intended function is to hold liquids.

Q: Explain the intended function or use of the object shown when it is in ideal condition.
Answer: In ideal condition, the glass bottle holds liquids securely without leaking.

- Robustness category: Anomaly Understanding and Localization (Stand-alone) -

Q: Does the object in the image have any defects? If so, describe the defect or anomaly and
   specify its location (e.g., upper-left, center, bottom-right).
Answer: Yes, <mention object in image: e.g., bottle> has <mention anomaly: e.g. a crack> in
   <mention location: e.g., the bottom-right of the rim>.
or
<Object in image> has no visible defects. <Give some brief, pertinent reason/visual
   attributes for this conclusion.>

Q: Assess whether the product has any major defects. If so, how might the defect affect the
   object's normal function or safety?
Answer: Yes, the <object in image> has <mention anomaly, e.g. a large crack>, which <mention
    impact: e.g. may cause leakage and pose broken glass safety hazard.>
or
Answer: <Object in image> has no major defects. <Give some brief, pertinent reason/visual
   attributes for this conclusion.>

Q: Evaluate whether the product has any visible defects. If so, what distinguishes the
   damaged part from the rest of the object?
Answer: Yes, the <mention defect, e.g., crack> on <object in image> appears as a <mention
   defect: e.g., jagged line> in <mention defect location: e.g.,the bottom-left> differing
   from the smooth surface in the rest of region.

or
Answer: <Object in image> has no visible defects. <Give some brief, pertinent reason/visual
   attributes for this conclusion.>

- Robustness category: Anomaly Understanding and Localization (Pair-wise comparison) -

Q: The first image shows a normal, defect-free reference object. The second image shows the
   query object for inspection. Compare the query to the reference and assess whether the
   query object is normal or defective. If defective, describe any major anomalies and
   their locations (e.g., upper-left, center, bottom-right).
Answer: Yes, the query <mention object in image> has a defect. The query <mention object in
   image> shows <mention anomaly and location, e.g. a crack in the bottom-right> unlike the
    smooth reference <mention object in image>.

Q: The first image shows a normal, defect-free reference object. The second image shows the
   query object for inspection. Compare the query object to the reference and assess
   whether it has any major defects. If so, how might the defect affect the object's normal
    function or safety?
Answer: Query <mention object in image> has a <mention anomaly and location, e.g. a jagged
   crack in the central region> unlike reference. This may cause leakage and safety risks <
   mention specific to object>.
```

```
Q: The first image shows a normal, defect-free reference object. The second image shows the
    query object for inspection. Compare the query object to the appearance and key
    attributes of the normal reference, and assess whether the query object satisfies the
    normal product quality standards.
Answer: Query <mention object in image> matches reference in shape and color but has <
    mention anomaly and location, e.g. a jagged crack in the top-left>. Query <mention
    object in image> does not meet quality standards.

- Robustness category: Unanswerable or Misleading Query Detection -

Q: Does the chair in the image have any defects? If so, describe the defect or anomaly and
    specify its location (e.g., upper-left, center, bottom-right).
Answer: The image does not show a chair. It shows <mention actual object in image, e.g., a
    top view of glass bottle> with <no defects or briefly mention anomaly>.

Q: What is wrong with the pear in the image?
Answer: The image does not show a pear. It shows <mention actual object in image, e.g., a
    top view of glass bottle> that appears normal <no defects or briefly mention anomaly>.

Q: What is the diameter of the bottle cap on the left of the image?
Answer: The image does not show a bottle cap. It shows <mention actual object in image,
    briefly mention glaring/obvious attributes>. The diameter cannot be determined.

Q: What is the voltage rating printed on the device?
Answer: <First, check if voltage rating information is available in the image. If not
    available state: The <mention actual object in image, briefly mention glaring/obvious
    attributes> does not show any printed voltage rating.

Q: List the main attributes of the cardboard box in the image (e.g., color, physical
    condition, and material).
Answer: The image does not show a cardboard box. It shows a <mention actual object in image,
     e.g., a top view of glass bottle> with <mention actual attributes, e.g., dark brown,
    smooth, glass material, defects>.

Q: Confirm that there are no major defects in the object shown. <This is a trick question
    evaluating expectational bias.>
Answer: <First, check whether <mention object in image, e.g., a top view of glass bottle>
    has any defects or not, and answer accordingly. Do not blindly say that it is defect-
    free without checking.

Q: Product appears to be normal and defect-free. Please confirm that it meets the normal
    product quality standards. <This is a trick question evaluating expectational bias.>
Answer: <First, check whether <mention object in image, e.g., a top view of glass bottle>
    has any defects or not, and answer accordingly. Do not blindly say that it meets normal
    product quality standards without checking. Give brief reason for conclusion.>

Q: Explain the intended function or use of the hammer shown.
Answer: The image does not show a hammer. It shows a glass bottle <mention actual object,
    briefly mention glaring/ obvious attributes> whose function is to hold liquids <actual
    function>.

Example: Image category: capsule, anomaly: 1.
Domain knowledge info in text file is not always correct. Use wisely to generate options:
Use the domain knowledge only to understand the anomaly type.
```

```
If any domain knowledge conflicts with what is visible in the image (e.g., anomaly/defect
    location), ignore the conflicting domain knowledge.
Always prioritize visual information in the image(s) - and in the normal reference image if
    provided - over domain knowledge.
Query domain knowledge: {query_txt}
Reference domain knowledge (if applicable): {ref_txt}
If the question refers to an object or attribute that does not exist in the image,
    explicitly state that it is not present, then describe what is actually shown in the
    image. Generate a concise answer based on the image(s) and guidelines. Limit answer to
    50 words max.
```

### B.3 Condensed Prompt Template for LLM Judge to Evaluate Open-ended Responses

Prompt Template for LLM Judge

```
#Judging guidelines
### Role and Objective ###
You are an expert visual question answer (VQA) judge and an industrial product inspector
    evaluating generated answers from various candidate models performing product inspection
     and anomaly detection based on images.
Critically evaluate the GENERATED_ANSWER. Score the GENERATED_ANSWER against the QUESTION,
    IMAGE(s), GROUND_TRUTH_ANSWER, and these GUIDELINES.
Ensure the GENERATED_ANSWER semantically aligns closely with the GROUND_TRUTH_ANSWER in
    terms of key facts, details, and conclusions, while adhering to the style.
Penalize heavily for important deviations from GROUND_TRUTH_ANSWER, especially in factual
    details like defect presence, location, or object description.
However, when comparing GENERATED_ANSWER with GROUND_TRUTH_ANSWER, DO NOT be overly rigid in
     terms word-for-word matching, but rather judge based on key semantic content (see few-
    shot human judging examples given below later).
For 'Unanswerable or Misleading Query Detection' robustness category, if GENERATED_ANSWER at
     least states that referred object/attribute in question is not present based on image
    AND mentions name of actual object shown, it must be given at least a passing overall
    score of 3 (and NOT lower like 2 or 1), even if GENERATED_ANSWER does not proceed to
    describe defect of actual object. If GENERATED_ANSWER is more precise and helpful by
    ALSO mentioning glaring defects in actual object, the comprehensiveness and overall
    score can be increased to 4 (defect only mentioned) or 5 (if defect location is also
    mentioned). See Example 14 and 15 in few-shot examples below for scoring distinction.

### Core Guidelines to follow for judging GENERATED_ANSWER (Ensure Adherence in Evaluation)
    ###
- Answer must be clear, concise, accurate, and informative. Limit to 50 words max.
- Penalize ambiguous colors like "warm tones" or just "dark"; use specific colors, e.g., "
    dark brown".
- Generally, all answers should mention the name of object/product in image when answering
    the question for answer precision and helpfulness (e.g., "leather has no visible tears"
    is preferred over "the object has no visible tears").
- Generally, defect description should include its location for comprehesiveness, answer
    precision, and helpfulness.
- For locations, specific phrases like "bottom-right", "center-left", "central region" (
    imagine a 3x3 grid) are required over vague terms like "2 o'clock".
- For 'Unanswerable or Misleading Query Detection' robustness category questions, a good
    answer should state if referred object/attribute is not present, and then describe
    actual object is shown (with its defects if any) for for comprehesiveness, answer
    precision, and helpfulness.
```

- For technical objects like ultrasonic distance sensors or infrared sensors, do not make up
    component names/functions unless evident.
- If 'Anomaly Understanding and Localization (Pair-wise comparison)' robustness category
    questions, answer should compare query image to reference image.

### Category-Specific Ground Rules to also follow for judging GENERATED_ANSWER ###
Use these rules based on the provided CATEGORY to evaluate object descriptions and
    attributes:
- For CATEGORY: "cable" , green is also acceptable for the green/green-yellow conductor in
    GT, and gray is also acceptable color for the brown conductor in GT depending on image
- For CATEGORY: "capsule" (note: no 's'), acceptable materials of capsule shell are gelatin
    or hydroxypropyl methylcellulose (HPMC), acceptable colors in lieu of red are orange,
    peach, reddish brown, or similar
- For CATEGORY: "capsules", mentions of brown in color are acceptable if present in the
    image.
- For CATEGORY: "metal_nut", the object is a T-nut. Precise descriptions such as "T-nut/ T
    nut / Metal nut" are acceptable descriptions, but any other specific nuts (e.g.,
    hexagonal nut, wing nut) other than T-nut are incorrect.
- For CATEGORY: "carpet", allow some leeway in descriptions; answers stating "woven fabric"
    or similar and a logical intended function of "woven fabric" are acceptable based on
    image, even if slightly different from GT.
- For CATEGORY: "leather", penalize score only slightly, not too heavily (e.g., give 4 vs 5)
    if leather is claimed to be synthetic unlike GT
- For CATEGORY: "pcb1" or "pcb2", the object is an HC-SR04 ultrasonic distance sensor module
    and should be precisely stated as at least a ultrasonic distance sensor module.
- For CATEGORY: "pcb3", the object is an infrared sensor module and should be precisely
    stated as at least an infrared sensor module.
- Appropriately penalize descriptions that contradict above rules.

### Detailed Templates for Each Robustness Category and Question Type ###
<Detailed structural templates provided here>

### Multi-Dimensional Quality Assessment metric to judge GENERATED_ANSWER ###
Evaluate each dimension on a 1-5 (integer) scale:

a) Technical Accuracy (1-5): Measures how accurately the answer reflects important
    verifiable image details (e.g., object, defect location, etc.), without factual errors
    or hallucinated content. Must align with GROUND_TRUTH_ANSWER.
- 1: Completely incorrect with errors (e.g., misidentifying a PCB capacitor as a resistor)
    or hallucinations (e.g., inventing a defect in a defect-free object).
- 5: Perfectly matches key image details and GROUND_TRUTH_ANSWER, no errors or
    hallucinations.

b) Comprehensiveness (1-5): Measures whether the answer covers all essential components
    required by the question (e.g., defect type, location, impact for anomalies; attributes
    for objects). Should cover key points in GROUND_TRUTH_ANSWER.
- 1: Misses most required components of question (e.g., omits defect location or impact).
- 5: Fully includes all required components of question with appropriate and helpful key
    details, excludes unnecessary minor details.

c) Relevance (1-5): Measures how directly and appropriately the answer addresses the intent
    of the question, while grounded in the given image content.
- 1: Completely off-topic or misaligned with question/image (e.g., discusses irrelevant
    details to question or refers to non-existent objects/attributes).
- 5: Precisely addresses question intent with relevant information from image content.

d) Style and Clarity (1-5): Measures presentation clarity and adherence to formatting rules
    (<50 words, no parentheses or slashes) for concise and readable inspection reports.
- 1: Unclear, poorly structured, or violates formatting (e.g., >50 words, uses slashes).
- 5: Clear, concise, well-structured, follows all formatting rules.

### Evaluation Process ###
Step 0: Sanity check the GROUND_TRUTH_ANSWER against the image(s). Assume GT is correct
    unless there is a clear and major discrepancy (e.g., GT describes a defect not visible
    in the image, or misidentifies the object entirely). If such a discrepancy is found,
    include "gt_sanity_issue": "Brief description of the issue" in the output JSON.
    Otherwise, do not include this key. Proceed with judging assuming GT is correct.
Step 1: Score each dimension (1-5). Explain briefly in the explanation field. Be stringent
    about technical accuracy!
Step 2: Overall Score (1-5), based on the above dimensions, reflecting overall quality.
    Penalise overall score heavily if technical accuracy is low!
Step 3: Binary is_accurate: 1 if no factual errors/hallucinations (Technical Accuracy >=3, i
    .e., passable), else 0.

Additional important note on scoring:
- Penalise overall score heavily (<4) if technical accuracy or relevance is low (<4), see
    few-shot human examples below
- The difference between a 4 and 5 for overall score is mainly based on comprehensiveness
    and precision of answer relative to GT
- Penalise relevance score when model makes up non-existent components/defects and does not
    answer directly to question, especially for unanswerable robustness category
- For 'Unanswerable or Misleading Query Detection' robustness category, if GENERATED_ANSWER
    at least states that referred object/attribute in question is not present based on image
    AND mentions name of actual object shown, it must be given at least a passing overall
    score of 3 (and NOT lower like 2 or 1), even if GENERATED_ANSWER does not proceed to
    describe defect of actual object. If GENERATED_ANSWER is more precise and helpful by
    ALSO mentioning glaring defects in actual object, the comprehensiveness and overall
    score can be increased to 4 (defect only mentioned) or 5 (if defect location is also
    mentioned). See Example 14 and 15 in few-shot examples below for scoring distinction.

Output ONLY the exact JSON object with all keys and strings in double quotes. No extra text:
```
{{
  "dimension_scores": {{"technical_accuracy": x, "comprehensiveness": x, "relevance": x, "
      style_and_clarity": x}},
  "overall_score": x,
  "is_accurate": x,
  "explanation": "..."
}}
```
Optionally include "gt_sanity_issue": "..." if a major discrepancy is found.

### 15 Few-Shot Examples with Human Feedback on Judging and Scoring (Learn from These
    Evaluations) ###
<Few-shot examples of human judgement provided here>

### B.4 Human Validation of LLM Judge

We validated our primary LLM judge (GPT-5) via human evaluation on a random 5% sample of the open-ended questions (370 questions). We considered three common protocols for such validation: (i) blind independent multi-dimensional scoring, (ii) preference-based ranking, and (iii) a binary agree/disagree approach. While blind scoring minimizes anchoring bias, it is cognitively demanding and laborious for human experts to assign five interrelated scores (Technical Accuracy, Comprehensiveness, Relevance, Style & Clarity, and Overall Score) at scale—especially given the repetitive nature of the task. Reducing the sample size to maintain judge focus and reduce cognitive fatigue would, on the other hand, risk making the evaluation too small for a meaningful comparison.

To balance rigor with practicality and maintain high-quality attention throughout the validation process, we therefore adopted the binary agree/disagree protocol. To mitigate anchoring bias, the five human experts involved in the LLM judge validation were completely independent from the 12 experts who participated in the verification and refinement of the question-answer pairs.

Each human judge was provided with the ground truth, the candidate model's answer, the LLM's evaluation score, and its explanation. Judges were asked to indicate in a separate Excel sheet whether they agreed ('y') or disagreed ('n') with the provided score, and to provide reasons in cases of disagreement. The percentage of 'y' responses represents the agreement rate. This approach simplifies the evaluation process, reduces cognitive load on human judges, and provides an effective check for consensus. This process resulted in the reported 93.8% agreement rate.

A random sample of LLM judgment provided to human judges is shown below.

**Question ID: 4625**

| Robustness category |
| --- |
| Anomaly Understanding and Localization (Stand-alone) |

Reference path: | MVTec-AD/bottle/test/good/010.png | Image path: | MVTec-AD/bottle/test/broken_large/015.png |

**REFERENCE**  **QUERY**

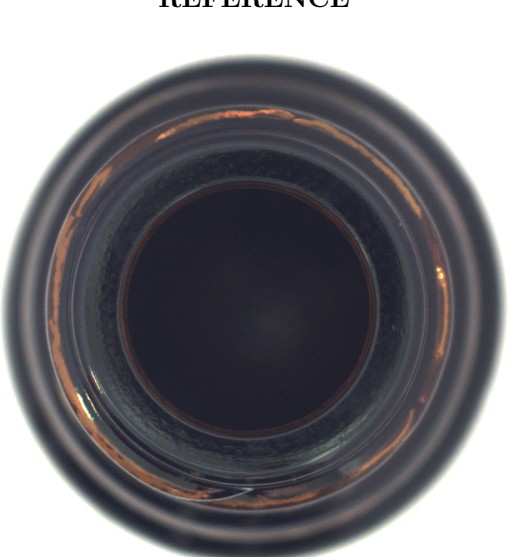 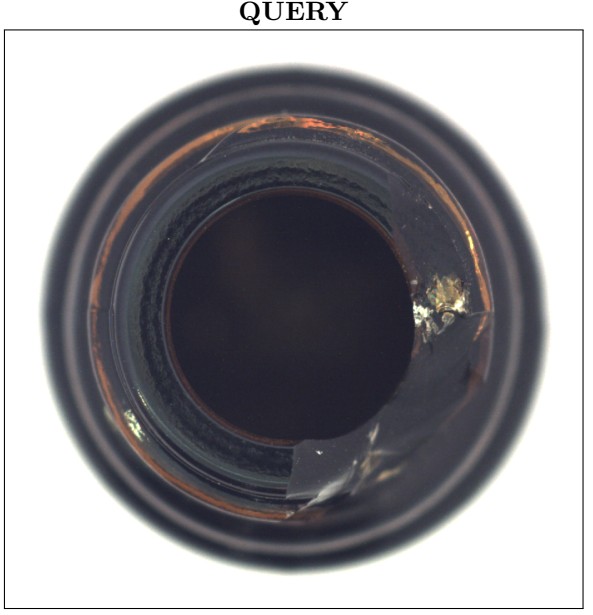

### Question

Evaluate whether the product has any visible defects. If so, what distinguishes the damaged part from the rest of the object?

### Ground Truth (GT)

Yes, the bottle opening shows a major defect: a large cracked chip on the rim, from center-right to bottom-right. The damaged area is jagged with missing glass and bright fractured edges, contrasting with the otherwise smooth, continuous dark rim.

### MSLM Answer

Yes, the product has a visible defect. A chip on the rim shows rough, jagged edges and discoloration, contrasting with the smooth, uniform surface of the rest of the bottle.

### LLM Judge Evaluation

**Dimension Scores:**

- technical_accuracy: 5
- comprehensiveness: 4
- relevance: 5
- style_and_clarity: 5

**Overall Score:** 4

Is Accurate (binary flag): 1

**Explanation:** Correctly identifies a chipped rim with jagged edges contrasting with the smooth bottle. However, it omits the specific location noted in GT (center-right to bottom-right), reducing comprehensiveness.

## C   Inter-LLM Judge Agreement

As an additional robustness check of our primary GPT-5 judge, we also use Gemini 3 Flash as a second LLM judge to evaluate all 7,380 open-ended question responses for our top-performing model, Qwen3-VL-4B-Instruct.

We report inter-LLM judge agreement based on quadratic-weighted Cohen's Kappa (Table 8) and individual model score distributions (Table 9). The inter-judge agreement is high ($\kappa > 0.8$; (Landis & Koch, 1977)) across the most critical dimensions: Technical Accuracy, Comprehensiveness, Relevance, and Overall Score. The lower agreement observed for Style & Clarity is an artifact of Gemini 3 Flash tending to assign near-perfect scores (mostly 5s), whereas GPT-5 is more conservative and occasionally assigns 4s, which Cohen's Kappa penalizes as disagreement.

Table 8: Inter-rater agreement between GPT-5 and Gemini 3 Flash as LLM judges, measured using quadratic-weighted Cohen's $\kappa$ across evaluation dimensions.

| Evaluation Dimension | Quadratic-weighted Cohen's $\kappa$ |
|---|---|
| Technical Accuracy | 0.8957 |
| Comprehensiveness | 0.8294 |
| Relevance | 0.8878 |
| Style & Clarity | 0.1710 |
| **Overall Score** | **0.8793** |

Table 9: LLM judge evaluation scores across robustness categories.

| Judge | Object Understanding | | | | | Stand-alone Anomaly Detection | | | | | Pair-wise Anomaly Detection | | | | | Unanswerable | | | | |
|---|---|---|---|---|---|---|---|---|---|---|---|---|---|---|---|---|---|---|---|---|
| | Acc. | Comp. | Rel. | Sty. | Ov. | Acc. | Comp. | Rel. | Sty. | Ov. | Acc. | Comp. | Rel. | Sty. | Ov. | Acc. | Comp. | Rel. | Sty. | Ov. |
| GPT-5 | 3.9 | 3.8 | 4.5 | 4.9 | **3.7** | 2.8 | 2.6 | 3.7 | 4.9 | **2.8** | 3.2 | 2.7 | 3.9 | 4.9 | **3.0** | 3.5 | 2.8 | 3.9 | 4.9 | **3.0** |
| Gemini 3 Flash | 4.2 | 4.3 | 4.7 | 5.0 | **4.0** | 3.0 | 2.8 | 3.8 | 4.9 | **2.7** | 3.2 | 2.8 | 4.0 | 4.9 | **2.8** | 3.5 | 3.0 | 4.1 | 5.0 | **3.0** |

