# OpenReview forum: "RobustMAD: Evaluating Real-World Robustness of Multimodal Small Language Models for Deployable Anomaly Detection Assistants"
_TMLR — Accepted by TMLR_

### Review · Reviewer_Ewao · 2026-04-27

**Summary Of Contributions:**

In this paper, the author(s) propose RobustMAD, a benchmark dataset for evaluating the usability/robustness of small multimodal language models against industrial anomaly detection (IAD). Existing work in this space has focused on either larger multi-modal models, which face cost/security issues in deployment vs. on-device smaller models, or do not consider domains outside of text-only. RobustMAD consists of 410 images sampled from two existing datasets (MVTec AD, and VisA), paired with over 10,000 MCQ/open-ended questions generated via GPT-5 to assess the ability of small multimodal models in several performance categories defined by the authors. In experiments, the authors find that while on-device models sometimes even outperforms closed-source models like GPT-nano, no models achieve strong overall performance.

**Additional Comments:**

- In general, the metrics/decisions used in the paper appear to be driven from ML research, rather than from the actual stakeholders in IAD. I am curious whether the authors think the research questions/results would be any different if an IAD expert was involved with the design of the benchmark?

**Audience:**

Yes

**Audience Explanation:**

Yes. I believe research into measuring the performance of on-device models in domain-specific fields (like IAD) lacks concrete evaluation methodologies, which this paper provides. I would be interested to know whether the author(s) can propose some concrete next steps/future work for newer MSLMs.

**Broader Impact Concerns:**

None.

**Claims And Evidence:**

Yes

**Claims Explanation:**

I think that the evidence presented by the authors that current MSLMs are not very performant in these tasks is clear and convincing. I have some concerns about the usage of LLMs as the judge for open-ended question responses, more details in requested changes/additional comments.

**Requested Changes:**

I have several questions about the design/evaluation of RobustMAD that I would like to see addressed in the paper:

1. During the question generation phase, it is stated that 560 hours was devoted to dataset refinement. Does that mean every question generated by GPT-5 was verified by an expert?
2. Why were only 5 objects selected from VisA? Were the other 7 objects in VisA dataset overlapping with MVTec?
3. Is there a sense of how comprehensive these 20 total objects are? For someone like not who is not familiar with IAD, 20 objects seems very limited in the grand scheme of assessing out-of-the-box performance of these models against IAD in general.
4. I was thinking more about the difference between technical accuracy and comprehensiveness, and I don’t believe there is a big enough difference to separate them. Shouldn’t a technically accurate answer also be comprehensive enough? For an expert in IAD, what kind of response would they prefer? Maybe the categories for evaluation should be grounded in that.
5. Can the authors provide Cohen Kappa alignment scores between the LLM judge and humans
6. Can the authors provide a citation for 4 out of 5 being a reasonable measure of “good-enough accuracy” in industry?

---

> ### Author Response · Authors · 2026-05-06
> **Author Response to Reviewer Ewao (1/5)**
>
> We sincerely thank the reviewer for the positive assessment of our work and for the constructive feedback provided. We have carefully addressed all concerns raised and provide below the point-by-point responses along with the corresponding manuscript revisions.
>
> **Response to Requested Change 1:**
>
> We thank the reviewer for this important clarifying question. Yes, every question–answer pair generated by GPT-5 underwent human expert verification as part of our two-stage quality control process.
>
> As described in Section 3.2 (RobustMAD Dataset Construction Pipeline), the draft dataset was first screened by GPT-5 to identify potentially problematic cases (e.g., technically inaccurate open-ended answers or ambiguous MCQ options). This was followed by extensive human review, with approximately 560 person-hours invested by 12 graduate student-level experts. During this stage, the experts carefully examined and refined all questions, including those that were not flagged by the initial GPT-5 screening.
>
> Accordingly, in the **revised manuscript**, we have clarified in Section 3.2 (RobustMAD Dataset Construction Pipeline) that this two-stage verification process applies to all questions.
>
> **Response to Requested Changes 2 and 3:**
>
> We thank the reviewer for the important questions on why only 5 objects were selected from the VisA dataset and whether the resulting 20 object types provide sufficient coverage for assessing the real-world robustness of MSLMs in industrial anomaly detection. We address these two closely related points together.
>
> **On the selection of only 5 objects from VisA**
>
> In constructing RobustMAD, we aimed to maximize diagnostic power and diversity of the dataset while maintaining the ability to perform rigorous human verification of every question-answer pair. The MVTec AD dataset already provides a strong foundation with 15 diverse single-object types. As a strategic complement, we selected 5 additional object types from VisA that meaningfully expand coverage in three critical dimensions underrepresented in MVTec AD: multi-object scenes, domain-knowledge-intensive objects, and objects with challenging fine-grained defects that better probe multimodal grounding. The selected objects were therefore candle, capsules, and three PCB variants (pcb1, pcb2, pcb3).
>
> The excluded VisA objects, primarily an additional PCB and several food-related items (e.g., macaroni, cashew), provide limited additional diversity in defect types and visual characteristics. Including these objects would substantially increase the human expert review burden (already approximately 560 person-hours across 12 experts) without meaningfully enhancing the benchmark’s diversity or diagnostic ability.
>
> In the **revised manuscript**, we have better clarified the rationale for VisA object selection in Section 3.3 (Data Statistics).
>
>
> **On whether 20 object types provide sufficient coverage**
>
> While 20 object types may appear modest at first glance, RobustMAD is intentionally designed with data efficiency and diagnostic depth in mind rather than sheer scale alone. We utilize 410 carefully selected representative images from the MVTec AD and VisA datasets, spanning 20 object types and 39 unique object-condition categories (including multi-defect scenarios). From these images, we meticulously curate 4,510 multiple-choice and 7,380 open-ended questions across four challenging knowledge-based robustness categories—General Object Understanding, Stand-alone Anomaly Detection, Pair-wise Anomaly Detection, and Unanswerable or Ill-posed Query Detection—that collectively capture the diverse and practical demands of real-world industrial anomaly inspection.
>
> Particularly, the benchmark provides **strategic and dense coverage** of critical challenges, including domain-knowledge-intensive scientific objects, fine-grained defects, cross-image reasoning, open-ended inspection queries, and notably unanswerable or ill-posed problems. The same questions are further evaluated under realistic visual quality degradations such as motion blur and low lighting. This focused yet comprehensive design enables, for the first time, a rigorous evaluation of the fragilities and failure modes of modern MSLMs that standard benchmarks often overlook.
>
> Our findings with RobustMAD successfully uncover significant robustness gaps and recurring failure modes in state-of-the-art MSLMs, including fragile multimodal grounding under fine-grained distinctions or degraded conditions, insufficiently comprehensive responses, and weak logical grounding on unanswerable queries. We believe these results demonstrate the benchmark’s effectiveness as a powerful diagnostic tool for guiding the design of next-generation multimodal anomaly detection assistants.
>
> We hope this satisfactorily addresses the reviewer’s concern regarding the coverage and representativeness of the dataset.

---

> ### Author Response · Authors · 2026-05-06
> **Author Response to Reviewer Ewao (2/5)**
>
> **Response to Requested Change 4:**
>
> We thank the reviewer for this thoughtful question, which highlights an important aspect of our evaluation design.
>
> Technical Accuracy and Comprehensiveness are complementary yet distinct dimensions that together enable holistic assessment of inspection-style reporting.
>
> - *Technical Accuracy (1-5)* measures how faithfully the response reflects verifiable image content (e.g., correct object name, defect type, and location) **without factual errors or hallucinations**.
> - *Comprehensiveness (1-5)* measures whether the response covers all essential components demanded by the query (e.g., object name + defect type + precise location) **with sufficient detail and specificity**.
>
> Technical accuracy targets outright factual mistakes, while comprehensiveness penalizes imprecise, vague, or underspecified answers that are technically not false but still inadequate for industrial use.
>
> This distinction is illustrated with representative examples from MiniCPM-V 4.0 responses in Figure 6e.
>
> - In the left example (PCB), the model responds: "The main attributes of the product in the image are: blue color, metallic pins on one end, a small integrated circuit chip, an LED light, and a black cylindrical component..."
>
> - In the right example (zipper), the model responds: "There is visible fraying and loose threads on the fabric near the zipper teeth, indicating wear or damage to the material around the zipper area..."
>
> While neither response contains outright factual errors (technical accuracy), the first answer is imprecise: it omits the actual product name and fails to provide precise location information. The second answer is underspecified: it correctly identifies fabric fraying but does not localize it precisely (center-right of the zipper teeth) and completely misses the second defect (broken zipper teeth in the top region). These responses are therefore penalized under the comprehensiveness dimension.
>
> **Literature-Based Grounding for Our Multi-Dimensional Scoring Criteria**
>
> We also clarify that the multi-dimensional approach we adopted is grounded in prior work on human-centered evaluation of real-world LLM capabilities (Miller & Tang, 2025; Liu et al., 2024), which advocate for scoring dimensions beyond just technical accuracy (e.g., relevance, clarity). In particular, in high-stakes domains such as clinical reporting, comprehensiveness is recognized as a critical criterion when evaluating LLM-generated reports (Lei et al., 2026). We draw similar motivation for industrial anomaly detection, where comprehensiveness is especially important because imprecise or incomplete reporting can lead to operational risks, particularly when multiple defects are present.
>
> An additional important benefit of our multi-dimensional criteria is that it provides *greater interpretability* of the overall scores. Human inspectors or model developers can easily identify which specific dimension (e.g., technical accuracy vs. comprehensiveness) caused a low overall score. This level of granularity cannot be achieved with a single aggregate accuracy metric alone.
>
> We hope this clarifies the reviewer’s concern and explains the value of including the additional comprehensiveness dimension for IAD inspection reporting. In the **revised manuscript**, we have clarified the Comprehensiveness criterion in Section 4.2 (Evaluation Setup and Metrics) to explicitly include the need for “precise descriptions” of all relevant details. We have also added the Lei et al. (2026) reference to further justify the distinction between technical accuracy and comprehensiveness.

---

> ### Author Response · Authors · 2026-05-06
> **Author Response to Reviewer Ewao (3/5)**
>
> **Response to Requested Change 5:**
>
> We thank the reviewer for raising this important question on the Cohen Kappa alignment. We fully agree with the importance of ensuring that the LLM judgment is robust, accurate, and well-aligned with human judgment.
>
> To recap, we validated our primary LLM judge (GPT-5) via human evaluation on a random 5% sample of the open-ended questions. We considered three common protocols for such validation: (i) blind independent multi-dimensional scoring, (ii) preference-based ranking, and (iii) a binary agree/disagree approach. While blind scoring minimizes anchoring bias, it is cognitively demanding and laborious for human experts to assign five interrelated scores (Technical Accuracy, Comprehensiveness, Relevance, Style & Clarity, and Overall Score) at scale—especially given the repetitive nature of the task. Reducing the sample size to maintain judge focus and reduce cognitive fatigue would, on the other hand, risk making the evaluation too small for a meaningful comparison.
>
> To balance rigor with practicality and maintain high-quality attention throughout the validation process, we therefore adopted the binary agree/disagree protocol. To mitigate anchoring bias, the five human experts involved in the LLM judge validation were completely independent from the 12 experts who participated in the verification and refinement of the question-answer pairs. These five experts were asked to carefully review the multi-dimensional scores and overall score assigned by the GPT-5 judge and indicate whether they agree or disagree. This process resulted in the reported 93.8% agreement rate.
>
> Although we cannot calculate Cohen’s Kappa between the LLM judge and human validators under this binary protocol, we agree that additional validation would further strengthen our LLM judge evaluations. Therefore, as an additional validation check, we have now tested Gemini 3 Flash as a second LLM judge on all 7,380 open-ended questions for our top-performing model, Qwen3-VL-4B-Instruct. We then computed quadratic-weighted Cohen’s Kappa (recommended for ordinal 1–5 scores) between the GPT-5 and Gemini 3 Flash scores. The results are as follows:
>
> | Evaluation Dimension   | Quadratic-weighted Cohen’s κ |
> |------------------------|------------------------------|
> | Technical Accuracy     | 0.8957                       |
> | Comprehensiveness      | 0.8294                       |
> | Relevance              | 0.8878                       |
> | Style & Clarity      | 0.1710                       |
> | **Overall Score**      | **0.8793**                   |
>
> The inter-judge agreement is high (κ > 0.8; Landis & Koch, 1977) on the most critical dimensions: Technical Accuracy, Comprehensiveness, Relevance, and Overall Score. The lower agreement on Style & Clarity is an artifact of Gemini 3 Flash tending to assign near-perfect scores (mostly 5s), while GPT-5 was more conservative with occasional 4s, which Cohen’s Kappa interprets as larger disagreement.
>
> In the **revised manuscript**, we have:
> -  Added a new Appendix C reporting the inter-LLM judge agreement (including Cohen’s Kappa values and score distributions) between our primary judge GPT-5 and the additional check Gemini 3 Flash.
> - Clarified the separation and independence between the human experts involved in dataset creation and those involved in LLM judge validation in Section 4.2 (Evaluation Setup and Metrics), thus clarifying the rigor of the human validation.
> - Discussed the trade-offs considered for different human validation protocols in Appendix B.4 (Human Validation of LLM Judge).
>
> We believe these additions meaningfully strengthen the reliability of our LLM judging framework. We are happy to provide further details or analyses if needed.
>
>
> **Response to Requested Change 6:**
>
> We thank the reviewer for this careful and valid observation. We agree that framing an average overall score of 4 out of 5 as “a reasonable minimum for stringent industrial standards” is subjective, especially given the nascent state of research in open-ended multimodal evaluation for industrial anomaly detection.
>
> Our intention behind highlighting this threshold was to emphasize that current top-performing MSLMs (e.g., Qwen3-VL-4B-Instruct achieving an average overall score below 4 in the open-ended setting) still fall short of the level of reliability, precision, and completeness typically expected in safety-critical industrial inspection scenarios. However, we acknowledge that a universal numerical threshold is difficult to justify without extensive domain-specific validation.
>
> In the **revised manuscript**, we have revised the relevant statement in the discussion of Table 6 of Section 4.3.2 (Open-ended Questions) to a more neutral and objective phrasing: “In contrast, Qwen3-VL-4B-Instruct now fails to exceed an average overall score of 4 (out of 5).”

---

> ### Author Response · Authors · 2026-05-06
> **Author Response to Reviewer Ewao (4/5)**
>
> **Response to Additional Comments:**
>
> > "In general, the metrics/decisions used in the paper appear to be driven from ML research, rather than from the actual stakeholders in IAD. I am curious whether the authors think the research questions/results would be any different if an IAD expert was involved with the design of the benchmark?"
>
> We thank the reviewer for this insightful comment regarding the extent to which our benchmark and metrics are grounded in the needs of actual IAD stakeholders. We fully agree that close alignment with domain experts is crucial for real-world impact.
>
> Multimodal visual assistants form a critical component of next-generation smart factories (Jiang et al., 2025). Prior work in this space has largely focused on ML research priorities: either evaluating large-scale MLLMs, which face prohibitive computational and privacy barriers for on-device use, or relying on restricted multiple-choice formats and image-only robustness tests (Pemula et al., 2025) that do not fully reflect realistic factory conditions.
>
> Motivated by the well-documented gap between controlled benchmark performance and real-world readiness in high-stakes domains (e.g., Gong et al., 2025 in healthcare), and recognizing that IAD faces similarly complex conditions—such as domain-intensive knowledge, dynamic image quality variations, ill-posed user queries, and the need for open-ended responses—we designed RobustMAD as a first step toward addressing these practical demands.
>
> Our work shifts multimodal IAD evaluation to be more meaningful for IAD inspectors (the primary stakeholders) in two key ways:
>
> - Moving away from controlled evaluation benchmarks with simple MCQs (which are unrealistic for daily inspection workflows) or rigid vision-only detection toward realistic and challenging open-ended querying and inspection reporting;
>
> - Focusing on multimodal small language models (MSLMs), which are far more amenable to on-device deployment in resource-constrained factory environments compared to large cloud-based MLLMs, while also mitigating privacy risks.
>
> For the dataset design, we built upon the broad anomaly inspection subtasks explored in MMAD (Jiang et al., 2025) but meaningfully extended them. We generated more realistic open-ended questions and introduced a new Unanswerable or Ill-posed Query Detection category. This addition is particularly important because real-world operators can ask non-standard queries, and the ability to remain logically grounded and avoid confident hallucinations is essential for safety-critical inspections.
>
> For the evaluation metrics, we designed a multi-dimensional, user-centered scoring scheme (technical accuracy, comprehensiveness, relevance, and style & clarity), drawing inspiration from human-centered LLM evaluation practices (Miller & Tang, 2025), including those in high-stakes domains such as clinical reporting (Lei et al., 2026).
>
> We believe RobustMAD provides a strong and useful foundation that reveals critical robustness gaps overlooked by existing benchmarks. Nevertheless, we fully agree that deeper involvement of IAD domain experts would further strengthen the benchmark. In the **revised manuscript**, we  explicitly highlight deeper collaboration with IAD stakeholders (e.g., incorporating expert feedback on queries and response preferences, validation of failure modes, and expansion of robustness categories based on real factory workflows) as an important direction for future work in Section 4.5 (Limitations).
>
> Thank you again for this valuable feedback, which helps us better articulate the domain relevance of our contributions.

---

> ### Author Response · Authors · 2026-05-06
> **Author Response to Reviewer Ewao (5/5)**
>
> **Response to Other Comments:**
> > "I would be interested to know whether the author(s) can propose some concrete next steps/future work for newer MSLMs."
>
> We thank the reviewer for raising this important point. Indeed, RobustMAD is designed to serve as a diagnostic benchmark that not only uncovers critical failure modes but also provides concrete guidance for the development of robust next-generation MSLMs.
>
> Our evaluation reveals recurring issues such as technical inaccuracies due to shallow domain grounding, missed fine-grained visual distinctions, vague or underspecified responses, and a lack of logically grounded reasoning when faced with unanswerable or ill-posed queries. These limitations cannot be fixed by prompt engineering alone and require targeted improvements in model architecture and training.
>
> Concrete next steps include:
> - **Vision-side architectural enhancements**: Careful design of the vision encoder, tight vision–language token alignment (e.g., DeepStack fusion), and explicit cross-image reasoning. This is evidenced by Qwen3-VL-4B-Instruct substantially outperforming InternVL3.5-4B despite sharing the same powerful Qwen3 language backbone.
>
> - **Targeted training data curation**: Explicitly address failure modes by training on anomaly knowledge, fine-grained visual distinctions, multi-defect scenarios, practical image quality degradations, and cross-image content differences.
>
> - **Explicit negative and counterfactual training**: Teach models to reliably handle unanswerable or ill-posed queries, as robustness in these settings does not emerge implicitly, even with the strong reasoning-oriented training of modern MSLMs. RobustMAD’s comprehensive robustness categories, including the explicit unanswerable category, human-verified ground truth labels, and standardized evaluation criteria for open-ended answers, provide a rigorous, human-verified template for training and assessing these robustness aspects.
>
> In the **revised manuscript**, these directions are discussed in detail in Section 4.4 (Lessons and Practical Guidance for Next-Generation Industrial Anomaly Inspection Assistants).

---

### Review · Reviewer_5gnA · 2026-04-28

**Summary Of Contributions:**

The paper introduces RobustMAD, a benchmark for evaluating multimodal small language models (MSLMs) in industrial anomaly-inspection tasks. It comprises 410 images, covers 20 object types and 39 object-condition categories. From these, it builds 4,510 multiple-choice questions and 7,380 open-ended questions for visual understanding, anomaly detection and unanswerable query handling. The same questions are evaluated under low-light and motion-blur perturbations. The authors evaluate several approximately 3B–8B models open-weight models, and proprietary reference models.

Results show that MSLMs are promising but still inadequate for industrial inspection. Qwen3-VL-4B-Instruct is the best of the small models, with a  88.31% aggregate MCQ accuracy.  The paper reports that it outperforms GPT-5 Nano on some aggregate RobustMAD metrics, while Gemini 3 Flash remains the top performing model The open-ended evaluation reveals much weaker performance than the MCQ setting, with Qwen3-VL-4B-Instruct reaching an average overall score of 3.12 out of 5 and 72.8% of responses at score ≥ 3.

**Audience:**

Yes

**Audience Explanation:**

Yes. It proposes a new diagnostic benchmark and surfaces failure modes for an interesting area. The unanswerable-query category is very valiable for the real world because it targets a failure mode that standard anomaly benchmarks and standard VQA datasets often miss. The paper brings a structured evaluation showing where current MSLMs fail and what future training/evaluation should target. That should be of interest to at least researchers working on multimodal robustness, small VLMs, VQA evaluation, and industrial AI.

**Broader Impact Concerns:**

Nothing to report.

**Claims And Evidence:**

Yes

**Claims Explanation:**

I believe the paper shows evidence of gaps in current MSLMs for industrial anomaly inspection. Benchmark categories are well motivated, evaluation covers both structured and open-ended formats, and the results reveal some trends: object understanding is easier than anomaly detection, domain-intensive objects are harder, pairwise references do not consistently solve anomaly detection, and open-ended inspection exposes weaknesses hidden by MCQs. The qualitative examples also strengthen the narrative by illustrating false negatives and hallucinations, and failures under degraded visual conditions.

However, some claims go slightly beyond the evidence. RobustMAD is framed as measuring “real-world robustness” and being “practically grounded,” but the benchmark is built from two public datasets, uses template/GPT-generated queries rather than real technician queries, and only considers fixed low-light and motion-blur perturbations. I would ask the authors to either add evidence from real inspection logs, user queries, or soften the framing to “deployment-inspired” or “real-world-motivated.”

The LLM-as-judge protocol is reasonable but needs stronger validation. The reported 93.8% agreement from five expert judges is useful, but humans appear to validate the LLM’s scores rather than independently score outputs blind, which may introduce anchoring bias. The authors should report independent human scoring on a stratified subset, inter-annotator agreement, correlation with GPT-5 scores, and agreement by dimension/category.

**Requested Changes:**

- Soften or better justify strong claims such as “real-world robustness,” “comprehensive,” “first,” and “far below industrial requirements.”
- Strengthen validation of the LLM-as-judge evaluation, ideally with blind human scoring on a subset and/or a second judge model.

---

> ### Author Response · Authors · 2026-05-06
> **Author Response to Reviewer 5gnA (1/2)**
>
> We sincerely thank the reviewer for the positive assessment of our work and for the constructive feedback provided. We have carefully addressed all concerns raised and provide below the point-by-point responses along with the corresponding manuscript revisions.
>
>
> **Response to Requested Change 1:**
>
> We thank the reviewer for this constructive feedback on the validity of our claims and for positively highlighting several strengths of our work.
>
> Multimodal visual assistants form a critical component of next-generation smart factories (Jiang et al., 2025). However, research in multimodal industrial anomaly detection remains nascent. Prior work has largely focused on evaluating large-scale MLLMs, which are often impractical for on-device deployment due to high computational cost and privacy concerns, or relied on restricted multiple-choice formats and image-only robustness tests (Pemula et al., 2025) that do not fully capture the complexities of real factory environments.
>
> Motivated by the well-documented gap between controlled benchmark performance and real-world robustness in high-stakes domains (e.g., Gong et al., 2025), and recognizing that industrial anomaly detection faces similar challenges—including domain-intensive knowledge, dynamic image quality variations, ill-posed user queries, and the need for open-ended responses—we designed RobustMAD as an important step toward more practical and deployment-relevant evaluation. Our goal is to push multimodal IAD evaluation closer to real-world readiness by emphasizing realistic open-ended querying under diverse and imperfect settings, and focusing on MSLMs that are required for practical deployment in resource-constrained factory environments.
>
> Nonetheless, we agree that some claims in the original manuscript may have been phrased too strongly, particularly as we have not yet incorporated feedback from real inspection logs or professional inspectors. In the **revised manuscript**, we have carefully softened and clarified the following claims:
>
> - Replaced “first practically grounded benchmark, designed to systematically evaluate real-world robustness” with “first deployment-motivated benchmark, designed to comprehensively evaluate model robustness through diverse open-ended queries spanning object understanding, anomaly detection, unanswerable problems, and visual quality degradations.” This reflects that RobustMAD is the first to comprehensively evaluate deployment-friendly MSLMs with open-ended queries, a dedicated unanswerable query category, cross-image reasoning, and visual quality robustness—aspects not jointly addressed in prior work such as MMAD (Jiang et al., 2025).
>
> - In the contributions list, we have qualified “first” with “to the best of our knowledge”.
>
> - Changed “far below industrial requirements” to “However, they still fall short of safety-critical requirements, and RobustMAD reveals critical robustness gaps that pose operational risks.” This maintains our core finding regarding the critical gaps in current MSLMs (as evidenced by low open-ended scores, fragile grounding, and hallucinations on unanswerable queries) while avoiding overly absolute language.
>
> - Retained the term “comprehensive” to describe RobustMAD’s broad coverage across four knowledge-based robustness categories plus visual quality robustness (evaluated on both multiple-choice and open-ended questions). This constitutes a significant expansion in both scope and depth of robustness evaluation relative to existing multimodal IAD benchmarks.
>
> - We have also explicitly highlighted deeper collaboration with IAD stakeholders (e.g., incorporating expert feedback on queries and response preferences, validation of failure modes, and expansion of robustness categories based on real factory workflows) as an important direction for future work in Section 4.5 (Limitations).
>
> These changes are applied consistently in the abstract, Section 1 (Introduction), the contributions list, Section 4.5 (Limitations), and Section 5 (Conclusion).
>
> We believe these revisions strike a better balance between conveying the motivation and contributions of our work while making the claims more precise and defensible.
>
> Thank you again for this helpful feedback, which has significantly improved the positioning and clarity of our work.

---

> ### Author Response · Authors · 2026-05-06
> **Author Response to Reviewer 5gnA (2/2)**
>
> **Response to Requested Change 2:**
>
> We thank the reviewer for this important suggestion to further strengthen the validation of our LLM-as-a-judge evaluation, ideally via blind human scoring on a subset and/or a second judge model.
>
> We fully agree with the importance of ensuring that the LLM judgment is robust, accurate, and well-aligned with human judgment.
>
> To recap, we validated our primary LLM judge (GPT-5) via human evaluation on a random 5% sample of the open-ended questions. We considered three common protocols for such validation: (i) blind independent multi-dimensional scoring, (ii) preference-based ranking, and (iii) a binary agree/disagree approach. While we acknowledge that blind scoring minimizes anchoring bias, it is cognitively demanding and laborious for human experts to assign five interrelated scores (Technical Accuracy, Comprehensiveness, Relevance, Style & Clarity, and Overall Score) at scale — especially given the repetitive nature of the task. Reducing the sample size to maintain judge focus and reduce cognitive fatigue would, on the other hand, risk making the evaluation too small for a meaningful comparison.
>
> To balance rigor with practicality and maintain high-quality attention throughout the validation process, we therefore adopted the binary agree/disagree protocol. To mitigate anchoring bias, the five human experts involved in the LLM judge validation were completely independent from the 12 experts who participated in the verification and refinement of the question-answer pairs. These five experts were asked to carefully review the multi-dimensional scores and overall score assigned by the GPT-5 judge and indicate whether they agree or disagree. This process resulted in the reported 93.8% agreement rate.
>
> Nonetheless, we agree that additional validation would further strengthen our LLM-as-a-judge evaluation. Therefore, following the reviewer’s suggestion, we have now tested Gemini 3 Flash as a second LLM judge on all 7,380 open-ended questions for our top-performing model, Qwen3-VL-4B-Instruct. We then computed quadratic-weighted Cohen’s κ (recommended for ordinal 1–5 scores) between the GPT-5 and Gemini 3 Flash scores. The results are as follows:
>
> | Evaluation Dimension   | Quadratic-weighted Cohen’s κ |
> |------------------------|------------------------------|
> | Technical Accuracy     | 0.8957                       |
> | Comprehensiveness      | 0.8294                       |
> | Relevance              | 0.8878                       |
> | Style & Clarity      | 0.1710                       |
> | **Overall Score**      | **0.8793**                   |
>
> The inter-judge agreement between the GPT-5 and Gemini 3 Flash is high (κ > 0.8; Landis & Koch, 1977) on the most critical dimensions: Technical Accuracy, Comprehensiveness, Relevance, and Overall Score. The lower agreement on Style & Clarity is an artifact of Gemini 3 Flash tending to assign near-perfect scores (mostly 5s), while GPT-5 was more conservative with occasional 4s, which Cohen’s Kappa interprets as larger disagreement.
>
> In the **revised manuscript**, we have:
>
> - Clarified the separation and independence between the human experts involved in dataset creation and those involved in LLM judge validation in Section 4.2 (Evaluation Setup and Metrics), thus clarifying the rigor of the human validation.
>
> - Discussed the trade-offs considered for different human validation protocols in Appendix B.4 (Human Validation of LLM Judge).
>
> - Added a new Appendix C reporting the inter-LLM judge agreement (including Cohen’s Kappa values and score distributions) between our primary judge GPT-5 and the additional check Gemini 3 Flash.
>
> We believe these additions meaningfully strengthen the reliability of our LLM judging framework. We are happy to provide further details or analyses if needed.

---

> > ### Comment · Reviewer_5gnA · 2026-06-04
> > **Acknowledgement**
> >
> > Thank you for your response. I am satisfied with the changes and update my recommendation to acceptance. The changes address my concerns by softening some claims and strengthening the LLM as a judge protocol. The benchmark appears useful and has meaningful use, which makes it relevant to the TMLR readers.

---

> > > ### Author Response · Authors · 2026-06-05
> > > **Author Acknowledgement**
> > >
> > > We sincerely thank the reviewer for taking the time to reassess our revised paper. We greatly appreciate your valuable comments throughout the review process, which helped strengthen our work. We are pleased that the revisions have addressed your concerns. Thank you again for your constructive feedback and support.

---

### Review · Reviewer_S3QJ · 2026-04-29

**Summary Of Contributions:**

This paper proposes RobustMAD, an evaluation framework for in industrial anomaly inspection targeting Multi-Modal small models. The proposed dataset includes 410 images samples from existing datasets, as well as 4,510 multiple-choice and 7,380 open-ended questions. The dataset covers four different categories.

**Audience:**

Yes

**Audience Explanation:**

Benchmarks for model evaluation in a real-world setting would be interesting to some of TMLR’s audience.

**Claims And Evidence:**

Yes

**Claims Explanation:**

The different categories of the benchmarks are well-motivated and fitting for the designed task.

On the evaluation side, the paper conducted comprehensive evaluation of the dataset using current small models. From the reported results, it seems like Qwen3-VL-4B was the best performing small model. On the other hand, Gemini 3 had the overall best performance. It would be interesting to see some analysis on why GPT-5 underperformed in many test areas.

More support maybe needed for the claim that “RobustMAD is the first practically grounded benchmark for evaluating the real-world robustness of MSLMs in industrial anomaly inspection.” The authors may want to include feedback from domain experts to avoid overclaiming.

**Requested Changes:**

Since the authors included the comparison, what are the key differences between a benchmark for small models versus a large language model? Is it realistic for a small model on a handheld device to perform well on all metrics?

Would be nice to see some more justification for openended questions with LLM-as-Judge.

---

> ### Author Response · Authors · 2026-05-06
> **Author Response to Reviewer S3QJ (1/4)**
>
> We sincerely thank the reviewer for the positive assessment of our work and for the constructive feedback provided. We have carefully addressed all concerns raised and provide below the point-by-point responses along with the corresponding manuscript revisions.
>
> **Response to Requested Change 1:**
>
> We thank the reviewer for the insightful questions regarding the comparison between benchmarks for small models and large language models, and whether it is realistic for small, resource-constrained models to perform well across all evaluation metrics. We address these closely related questions together for a complete picture.
>
> **Key differences between a benchmark for small models versus large models**
>
> We fully agree that the design considerations for an MSLM-targeted benchmark differ meaningfully from those for  MLLMs, and we made these considerations explicit in RobustMAD’s design. Three differences are central to our work:
>
> - **Deployment-relevant evaluation conditions.** MSLMs are designed for resource-constrained edge devices (e.g., factory handhelds, embedded inspection cameras), where operating conditions are far less controlled than in curated large-scale MLLM benchmarks. In practice, this includes realistic imperfect conditions such as low lighting and motion blur, which commonly occur in on-site industrial inspection workflows. RobustMAD therefore evaluates models on both clean images and these degradations (Tables 5 and 7), which are not jointly considered in existing IAD benchmarks targeting larger MLLMs (e.g., MMAD; Jiang et al., 2025).
>
> - **Diagnostic depth over sheer scale.** For MSLMs, the failure modes that matter for safe deployment—fragile fine-grained grounding, hallucinations on ill-posed queries, and underspecified reports—are easily masked by aggregate accuracy. RobustMAD therefore prioritizes diagnostic depth: 410 carefully selected images yield 4,510 MCQs and 7,380 open-ended questions structured around four robustness categories, each enabling targeted analysis of where models fail. *This same diagnostic design is challenging enough to surface meaningful gaps in larger reference MLLMs as well, not only in MSLMs.*
>
> - **Realistic open-ended querying with multi-dimensional scoring.** Practical inspection workflows naturally involve open-ended queries rather than multiple-choice questions, which are generally not readily available in real-world settings. RobustMAD provides a quantified and standardized evaluation framework for open-ended responses along four user-centered dimensions: Technical Accuracy, Comprehensiveness, Relevance, and Style & Clarity, ensuring that fluent but factually incorrect responses are not conflated with correct but underspecified ones (see Section 4.3.2 and Figure 5). Within this design, Gemini 3 Flash and the proprietary GPT-5 Nano are included as reference models rather than competitors on equal footing, to contextualize MSLM performance and indicate available headroom, while acknowledging that such models are not viable for on-device industrial deployment.
>
> **On whether small models can realistically perform well across all metrics**
>
> We clarify that RobustMAD is intentionally designed as a *diagnostic and aspirational* benchmark rather than one that current MSLMs are expected to fully solve. Industrial anomaly inspection is safety-critical: missed defects, vague reports, or confidently incorrect answers to ill-posed queries can cause costly failures. Therefore, a deployable inspection assistant must be near-perfect across all robustness categories. The fact that even the top-performing MSLM, Qwen3-VL-4B-Instruct, does not exceed an average overall open-ended score of 4 (out of 5) is precisely the gap RobustMAD is designed to expose.
>
> Our findings nonetheless suggest that this gap is not out of reach. Qwen3-VL-4B-Instruct outperforms the much larger Phi-4-Multimodal-Instruct (6B) and the proprietary GPT-5 Nano on overall MCQ accuracy and across most open-ended categories (Tables 3, 4, 6, and 7), indicating that strong robustness is achievable even at ~4B parameter scale. As discussed in Section 4.4, the remaining gap can be substantially reduced via (i) careful vision-side architectural choices such as DeepStack-style vision–language token alignment and explicit cross-image reference grounding, and (ii) targeted post-training that covers domain knowledge, fine-grained defects, multi-defect scenarios, image-quality degradations, and unanswerable queries using a deployment-motivated benchmark such as RobustMAD. These are tractable directions for the community rather than inherent limitations of small-model capacity.
>
> In the **revised manuscript**, design considerations for RobustMAD development and aspirational framing have been made more explicit in Sections 3 (Robust Multimodal Industrial Anomaly Detection Benchmark), 4.1 (Baselines), and 4.4 (Lessons and Practical Guidance for Next-Generation Industrial Anomaly Inspection Assistants).

---

> ### Author Response · Authors · 2026-05-06
> **Author Response to Reviewer S3QJ (2/4)**
>
> **Response to Requested Change 2:**
>
> We thank the reviewer for the valuable suggestion to provide additional justification for our LLM-as-a-judge evaluation protocol. The LLM-as-a-judge approach enables scalable evaluation of candidate model responses across multiple interrelated dimensions that are important in industrial inspection reporting (Technical Accuracy, Comprehensiveness, Relevance, and Style & Clarity). These dimensions would be difficult to evaluate at scale with human experts alone.
>
> We fully agree, however, that it is essential to validate that the LLM judgment is robust, accurate, and well-aligned with human expert judgment. To ensure our LLM judgment are aligned with human judgment, we validated our primary LLM judge (GPT-5) via human evaluation on a random 5% sample of the open-ended questions. We considered three common protocols for such validation: (i) blind independent multi-dimensional scoring, (ii) preference-based ranking, and (iii) a binary agree/disagree approach. While blind scoring may minimize anchoring bias, it is cognitively demanding and laborious for human experts to assign five interrelated scores (Technical Accuracy, Comprehensiveness, Relevance, Style & Clarity, and Overall Score) at scale—especially given the repetitive nature of the task.
>
> To balance rigor with practicality and maintain high-quality attention throughout the validation process, we therefore adopted the binary agree/disagree protocol. To mitigate anchoring bias, the five human experts involved in the LLM judge validation were completely independent from the 12 experts who participated in the verification and refinement of the question-answer pairs. These five experts were asked to carefully review the multi-dimensional scores and overall score assigned by the GPT-5 judge and indicate whether they agree or disagree. This process resulted in the reported 93.8% agreement rate.
>
> Nonetheless, we agree that additional validation would strengthen the justification for our LLM-as-a-judge approach. Therefore, as an additional robustness check, we have now used Gemini 3 Flash as a second LLM judge on all 7,380 open-ended questions for our top-performing model, Qwen3-VL-4B-Instruct. We then computed the inter-judge agreement between the GPT-5 and Gemini 3 Flash scores using the quadratic-weighted Cohen’s κ (recommended for ordinal 1–5 scores). The results are as follows:
>
> | Evaluation Dimension   | Quadratic-weighted Cohen’s κ |
> |------------------------|------------------------------|
> | Technical Accuracy     | 0.8957                       |
> | Comprehensiveness      | 0.8294                       |
> | Relevance              | 0.8878                       |
> | Style & Clarity      | 0.1710                       |
> | **Overall Score**      | **0.8793**                   |
>
> The inter-judge agreement between the GPT-5 and Gemini 3 Flash is high (κ > 0.8; Landis & Koch, 1977) on the most critical dimensions: Technical Accuracy, Comprehensiveness, Relevance, and Overall Score. The lower agreement on Style & Clarity is an artifact of Gemini 3 Flash tending to assign near-perfect scores (mostly 5s), while GPT-5 was more conservative with occasional 4s, which Cohen’s Kappa interprets as larger disagreement.
>
> Together with the human validation of the LLM judgments, we believe this provides additional justification for our current LLM-as-a-judge evaluation.
>
> In the **revised manuscript**, we have:
> - Clarified the separation and independence between the human experts involved in dataset creation and those involved in LLM judge validation in Section 4.2 (Evaluation Setup and Metrics), thus clarifying the rigor of the human validation.
> - Discussed the trade-offs considered for different human validation protocols in Appendix B.4 (Human Validation of LLM Judge).
> - Added a new Appendix C reporting the inter-LLM judge agreement (including Cohen’s Kappa values and score distributions) between our primary judge GPT-5 and the additional check Gemini 3 Flash.
>
> We believe these additions meaningfully strengthen the reliability of our LLM judging framework. We are happy to provide further details or analyses if needed.

---

> ### Author Response · Authors · 2026-05-06
> **Author Response to Reviewer S3QJ (3/4)**
>
> **Response to Other Comments:**
> > *"It would be interesting to see some analysis on why GPT-5 underperformed in many test areas."*
>
> We thank the reviewer for this insightful observation. The relative underperformance of the proprietary GPT-5 Nano on several RobustMAD categories is indeed a noteworthy finding of our evaluation, and we appreciate the opportunity to elaborate further.
>
> We first clarify that the proprietary OpenAI model included in our evaluation is GPT-5 Nano, the smallest and most distilled variant of the GPT-5 family, optimized for low-latency inference, rather than the full GPT-5 model. Its scale and intended use case make it a fairer reference point for comparison with MSLMs that target on-device deployment. In contrast, larger proprietary models, such as full GPT-5 or Gemini 3 Pro, are not representative of the deployment regime that motivates our work.
>
> To recap, GPT-5 Nano lags behind well-designed open-source MSLMs across several RobustMAD categories. In the multiple-choice setting (Table 3), it achieves 82.26% overall accuracy, compared to 88.31% for Qwen3-VL-4B-Instruct, with the largest gaps in Pair-wise Anomaly Detection (79.63% vs. 84.88%) and Unanswerable or Ill-posed Query Detection (76.34% vs. 86.02%). This pattern persists in the open-ended setting (Table 6), where GPT-5 Nano attains an average overall score of 2.73 versus 3.12 for Qwen3-VL-4B-Instruct, and remains consistent under low visual quality conditions (Table 7).
>
> We attribute this underperformance to three contributing factors, consistent with the recurring failure modes identified in Section 4.3.2 and with the practical guidance discussed in Section 4.4 :
>
> - **Limited fine-grained visual grounding.** GPT-5 Nano's relative weakness is most pronounced on tasks requiring precise visual discrimination, including fine-grained defect localization, pair-wise cross-image comparison, and image-grounded handling of ill-posed queries. This is consistent with the well-documented tendency of distilled or efficiency-oriented proprietary models to retain strong language priors while compressing vision-side capacity, which limits multimodal grounding on domain-specific images.
>
> - **Architectural choices that favor top-performing MSLMs.** Top MSLMs such as Qwen3-VL-4B-Instruct benefit from carefully designed vision-side mechanisms, particularly DeepStack-style fusion (Meng et al., 2024), which routes visual tokens from multiple encoder layers to corresponding LLM layers for tighter vision–language alignment. These design choices appear especially impactful in the categories where GPT-5 Nano lags. Importantly, this reinforces a key lesson from our results: thoughtful vision-side architectural design can outweigh raw model scaling for industrial anomaly inspection.
>
> - **Targeted training data exposure.** Modern MSLMs such as Qwen3-VL-4B-Instruct have been explicitly trained on synthetic omni-datasets covering diverse real-world scenarios and cross-image reasoning (Yang et al., 2025). Proprietary models may not necessarily be exposed to similarly targeted and curated coverage, which can disadvantage them on domain-specific benchmarks like RobustMAD.
>
> We view these findings as a strong demonstration of the diagnostic value of RobustMAD: it surfaces robustness gaps that are not apparent from model scale or general benchmark performance alone, and it highlights concrete, actionable directions, including vision-side architecture and targeted training data, for addressing these gaps.
>
> In the **revised manuscript**, we have added a brief discussion of GPT-5 Nano’s relative underperformance and its contributing factors to Section 4.4 (Lessons and Practical Guidance for Next-Generation Industrial Anomaly Inspection Assistants).

---

> ### Author Response · Authors · 2026-05-06
> **Author Response to Reviewer S3QJ (4/4)**
>
> > "More support maybe needed for the claim that “RobustMAD is the first practically grounded benchmark for evaluating the real-world robustness of MSLMs in industrial anomaly inspection.” The authors may want to include feedback from domain experts to avoid overclaiming."
>
> We thank the reviewer for this constructive feedback regarding the strength of our claims and for the valuable suggestion to incorporate domain expert perspectives.
>
> Multimodal visual assistants form a critical component of next-generation smart factories (Jiang et al., 2025). However, research in multimodal industrial anomaly detection remains nascent. Prior work has largely focused on evaluating large-scale MLLMs, which are often impractical for on-device deployment due to high computational cost and privacy concerns, or relied on restricted multiple-choice formats and image-only robustness tests (Pemula et al., 2025) that do not fully capture the complexities of real factory environments.
>
> Motivated by the well-documented gap between controlled benchmark performance and real-world robustness in high-stakes domains (e.g., Gong et al., 2025), and recognizing that industrial anomaly detection faces similar challenges—including domain-intensive knowledge, dynamic image quality variations, ill-posed user queries, and the need for open-ended responses—we designed RobustMAD as an important step toward more practical and deployment-relevant evaluation. Our goal is to push multimodal IAD evaluation closer to real-world readiness by emphasizing realistic open-ended querying under diverse and imperfect settings, and focusing on MSLMs that are required for practical deployment in resource-constrained factory environments.
>
> Nonetheless, we agree that some claims in the original manuscript may have been phrased too strongly, particularly as we have not yet incorporated feedback from real inspection logs or professional inspectors. In the **revised manuscript**, we have carefully softened and clarified the following claims:
>
> - Replaced “first practically grounded benchmark, designed to systematically evaluate real-world robustness” with “first deployment-motivated benchmark, designed to comprehensively evaluate model robustness through diverse open-ended queries spanning object understanding, anomaly detection, unanswerable problems, and visual quality degradations.” This reflects that RobustMAD is the first to comprehensively evaluate deployment-friendly MSLMs with open-ended queries, a dedicated unanswerable query category, cross-image reasoning, and visual quality robustness—aspects not jointly addressed in prior work such as MMAD (Jiang et al., 2025).
>
> - In the contributions list, we have qualified “first” with “to the best of our knowledge”.
>
> - We have also explicitly highlighted deeper collaboration with IAD experts (e.g., incorporating expert feedback on queries and response preferences, validation of failure modes, and expansion of robustness categories based on real factory workflows) as an important direction for future work in Section 4.5 (Limitations).
>
> These changes are applied consistently in the abstract, Section 1 (Introduction), the contributions list, Section 4.5 (Limitations), and Section 5 (Conclusion).
>
> We believe these revisions strike a better balance between conveying the motivation and contributions of our work while making the claims more precise and defensible.
>
> Thank you again for this helpful feedback, which has significantly improved the positioning and clarity of our work.

---

### Author Response · Authors · 2026-05-06
**Response and Revision Summary**

We sincerely thank the editors and reviewers for their time, effort, and valuable feedback on our manuscript. In response to these constructive comments, we have carefully addressed all concerns and revised the paper accordingly. The revised manuscript has been submitted, with all changes clearly highlighted in blue. Point-by-point responses are provided in our comments to the respective reviewers. For ease of reference, we provide a summary below:

**Summary of Key Contributions**

RobustMAD is the first deployment-motivated benchmark for evaluating the robustness of multimodal small language models (MSLMs) in industrial anomaly inspection. Leveraging 410 representative images from MVTec AD and VisA, we meticulously curate 4,510 multiple-choice and 7,380 open-ended questions spanning four challenging knowledge-based robustness categories and two types of deployment-relevant visual quality degradations (motion blur and low lighting). By systematically covering realistic and diverse challenges, such as domain knowledge–intensive scientific objects, fine-grained defects, visual quality perturbations, cross-image reasoning, open-ended queries, and notably unanswerable problems, RobustMAD enables, for the first time, a rigorous evaluation of the fragilities and failure modes of modern MSLMs. Under open-ended evaluation, three recurring failure modes emerge: (i) fragile multimodal grounding under fine-grained distinctions or degraded conditions, (ii) insufficiently comprehensive responses, and (iii) weak logical grounding on unanswerable queries leading to hallucinations. Our multi-dimensional user-centered scoring scheme (Technical Accuracy, Comprehensiveness, Relevance, and Style & Clarity) enables standardized quantitative assessment of open-ended responses with greater interpretability, allowing human inspectors and model developers to directly identify which dimensions drive low overall scores. Together, the RobustMAD benchmark and our findings offer a timely evaluation of current MSLM readiness, reveals critical failure modes posing significant operational risks, and provides actionable guidance for next-generation multimodal industrial inspection assistants.

**The major revisions are summarized as follows (in top-down order of reviewers as they appear):**

- We have discussed specific design considerations for deployment-motivated benchmarks (imperfect deployment conditions, diagnostic depth, and open-ended evaluation) in Section 3.
- We have added an analysis of GPT-5 Nano’s underperformance along with enriched practical guidance for developing next-generation MSLMs in Section 4.4.
- We have substantially strengthened the GPT-5-based LLM-as-a-judge evaluation by clarifying human validation expert independence in Section 4.2, discussing different human validation trade-offs in Appendix B.4, and adding Appendix C on inter-judge agreement (Cohen's Kappa) with another model, Gemini 3 Flash.
- Where appropriate, we have clarified how our benchmark pushes multimodal industrial anomaly detection towards real-world robustness and its value to IAD stakeholders. In the remaining instances, we have softened strong claims in the abstract, Introduction, contributions list, and Conclusion, while identifying areas for deeper future collaboration with IAD domain experts in Section 4.5.
- We have clarified the dataset design rationale in Section 3.3, justifying the strategic selection of VisA objects and highlighting dense diagnostic coverage.
- We have clarified the distinction and value of Technical Accuracy vs. Comprehensiveness and further grounded our multi-dimensional scoring in relevant literature on evaluating real-world capabilities of LLM-generated reports.

---

### Decision · Action_Editor_vWFk · 2026-06-11

**Recommendation:** Accept as is

**Audience:**

Yes

**Audience Explanation:**

Some individuals would be interested in the findings of the paper.

**Claims And Evidence:**

Yes

**Claims Explanation:**

Three expert reviewers reviewed the manuscript and gave a split set of recommendations. The main remaining concern, raised by one reviewer, is how much the evaluation focused on domain experts-vs-ML experts. The AE completely agrees with the reviewer that the paper would be improved with an evaluation that integrated domain experts deeply (and this could perhaps be a good direction for future research). However, the AE thinks that the depth of domain engagement in the evaluation is sufficient for a ML paper. Apart from this, the reviewers overall thought that the papers' claims were supported.

---

> ### Author Response · Authors · 2026-06-18
> **Author Acknowledgement**
>
> We sincerely thank the Action Editor and reviewers for their thoughtful evaluation and constructive feedback. We are grateful for the positive assessment of our work and for the decision to accept the manuscript.
>
> We appreciate the feedback regarding the potential benefits of deeper integration of domain experts in the evaluation process. We agree that this is a valuable direction for future research and will take this feedback into consideration in our future work.